# SenseFlow: Scaling Distribution Matching for Flow-based Text-to-Image Distillation

**Xingtong Ge**[1,2], **Xin Zhang**[3], **Tongda Xu**[4], **Yi Zhang**[3], **Xinjie Zhang**[1],
**Yan Wang**[4], **Jun Zhang**[1]*

[1]The Hong Kong University of Science and Technology, [2]SenseTime Research, [3]Vivix AI
[4]Institute for AI Industry Research, Tsinghua University
xingtong.ge@gmail.com, eejzhang@ust.hk

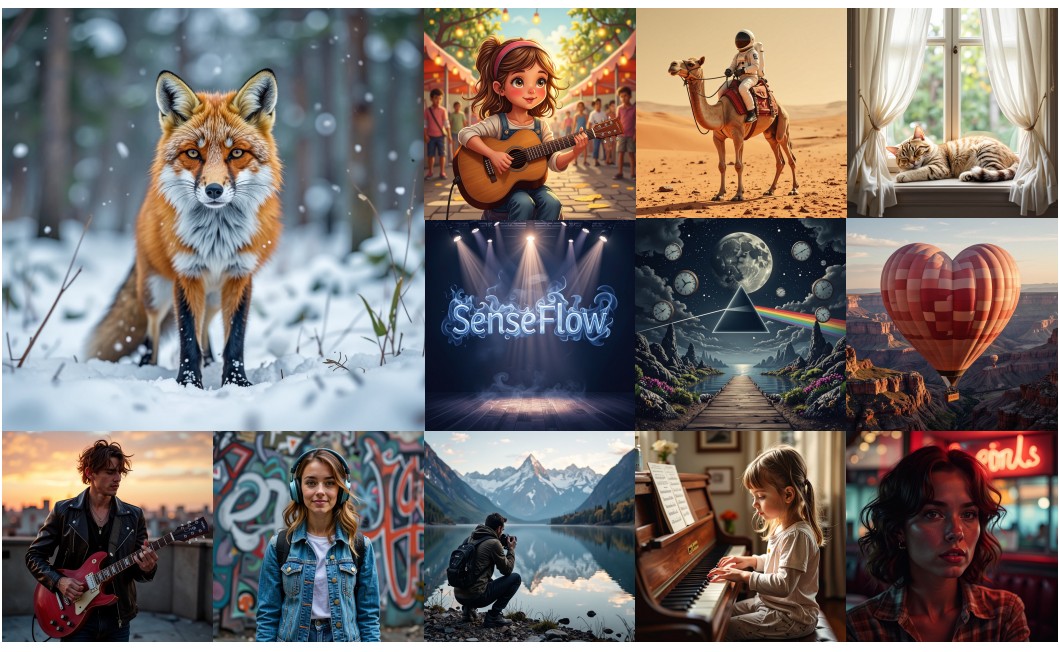

Figure 1: 1024×1024 samples produced by our 4-step generator distilled from FLUX.1-dev. Please zoom in for details.

## Abstract

The Distribution Matching Distillation (DMD) has been successfully applied to text-to-image diffusion models such as Stable Diffusion (SD) 1.5. However, vanilla DMD suffers from convergence difficulties on large-scale flow-based text-to-image models, such as SD 3.5 and FLUX. In this paper, we first analyze the issues when applying vanilla DMD on large-scale models. Then, to overcome the scalability challenge, we propose implicit distribution alignment (IDA) to constrain the divergence between the generator and the fake distribution. Furthermore, we propose intra-segment guidance (ISG) to relocate the timestep denoising importance from the teacher model. With IDA alone, DMD converges for SD 3.5; employing both IDA and ISG, DMD converges for SD 3.5 and FLUX.1 dev. Together with a scaled VFM-based discriminator, our final model, dubbed **SenseFlow**, achieves superior performance in distillation for both diffusion based text-to-image models such as SDXL, and flow-matching models such as SD 3.5 Large and FLUX.1 dev. The source code is available at https://github.com/XingtongGe/SenseFlow.

---

*Corresponding Author

# 1 INTRODUCTION

Significant advancements have been made on diffusion models (Ho et al., 2020; Rombach et al., 2022; Podell et al., 2024; Esser et al., 2024; Labs, 2024) for text-to-image generation over recent years. However, these models typically require multiple denoising steps to generate high-quality images. As models continue to scale up in terms of the parameter size, the computational cost and inference time for image generation increase substantially, making the process slower and more resource-intensive. To address this issue, various diffusion distillation methods have been developed to distill a diffusion model into a few-step generator, including consistency models (Song et al., 2023; Luo et al., 2023a; Wang et al., 2024a), progressive distillation (Salimans & Ho, 2022; Ren et al., 2024), adversarial distillation (Wang et al., 2023a; Sauer et al., 2024b; Chadebec et al.), and score distillation (Yin et al., 2024b;a; Luo et al., 2023b). Currently, the Distribution Matching Distillation (DMD) series (Yin et al., 2024a) have demonstrated superior results in distilling standard diffusion models such as SD 1.5 (Rombach et al., 2022) and SDXL (Podell et al., 2024).

However, few of these methods have successfully demonstrated effective distillation performance across a broader range of models, particularly in flow-based diffusion models with larger parameter sizes, such as SD3.5 Large (8B) (Esser et al., 2024) and FLUX.1 dev (12B) (Labs, 2024). As models increase in architecture complexity, parameter size, and training complexity, it becomes significantly more challenging to distill these models into efficient few-step generators (e.g., a 4-step generator).

In this paper, we introduce **SenseFlow**, which selects the framework of DMD2 (Yin et al., 2024a) as a touchstone, and scales it up for larger flow-based text-to-image models, including SD3.5 Large and FLUX.1 dev. Specifically, vanilla DMD2 has difficulty in converging and faces significant training instability on large models, even with the time-consuming two time-scale update rule (TTUR) (Heusel et al., 2017) applied. Viewing DMD as a min–max game, the *inner best response* requires the fake distribution model to track and predict the data distribution determined by generator samples effectively, which is brittle and expensive to realize. We therefore introduce *implicit distribution alignment (IDA)*: a lightweight proximal update applied after each generator step that nudges the fake model toward the generator, maintaining $p_f(x_t) \approx p_g(x_t)$—an $\varepsilon$-best response. This simple alignment markedly improves stability and enables convergence on large backbones.

Further, DMD2 and most existing diffusion distillation methods still use uniformly of handcrafted sampled timesteps for training and inference. However, due to the complex strategies employed during training of teacher diffusion models, different timesteps exert varying denoising effects throughout the entire process, which is also discussed in RayFlow (Shao et al., 2025). To avoid the inefficiency of naive timestep sampling strategy in distillation, we propose to *relocate* the teacher's timestep-wise denoising importance into a small set of selected coarse timesteps. For each coarse timestep $\tau_i$, we construct an *intra-segment guidance (ISG)* by sampling an intermediate timestep $t_{mid} \in (\tau_{i-1}, \tau_i)$, and construct a guidance trajectory: the teacher denoises from $\tau_i$ to $t_{mid}$, then the generator continues from $t_{mid}$ to $\tau_{i-1}$. We then guide the generator to align its direct prediction from $\tau_i$ to $\tau_{i-1}$ with this trajectory. This guidance mechanism effectively aggregates the teacher's fine-grained behavior within each segment, improving the generator's ability to approximate complex transitions across fixed sparse timesteps.

For further enhancement, we incorporate a more general and powerful discriminator built upon vision foundation models (*e.g.*, DINOv2 (Oquab et al., 2023), CLIP (Radford et al., 2021)), which operates in the image domain and can provide stronger semantic guidance. The use of pretrained vision backbones introduces rich semantic priors, enabling the discriminator to better capture image-level quality and fine-grained structures. By aggregating timestep-aware adversarial signals, this design yields stable and efficient training with superior visual qualities.

To summarize, we dive into the distribution matching distillation (DMD) and scale it up for a wide range of large-size flow-based text-to-image models. Our contributions are as follows:

- We discover that vanilla DMD2 suffers from the convergence issue on large-scale text-to-image models, even with TTUR applied. To tackle this challenge, we propose implicit distribution alignment to constrain the divergence between the generator and the fake model.
- To mitigate the problem of suboptimal sampling in DMD2, we propose intra-segment guidance to relocate the teacher's timestep-wise denoising importance, improving the generator's ability to approximate complex transitions across sparse timesteps.

- By incorporating a more powerful discriminator built upon vision foundation models with timestep-aware adversarial signals, we achieve stable training with superior performance.

- Experimental results show that our final model, dubbed **SenseFlow**, achieves state-of-the-art performance in distilling large-scale flow-matching models ( *e.g.*, SD 3.5, FLUX.1 dev) and diffusion-based models (*e.g.*, SDXL).

## 2 PRELIMINARIES

### 2.1 DIFFUSION MODELS

Diffusion models are a family of generative models, with the forward process perturbing the data $X_0 \sim p(X_0)$ to Gaussian noise $p(X_T) = \mathcal{N}(0, I)$ with a series distributions $p(X_t)$ defined by a forward stochastic differential equation (SDE):

$$dX_t = f(X_t, t)dt + g(t)dB_t, t \in [0, T] \tag{1}$$

where $f(X_t, t)$ is drifting parameter, $g(t)$ is diffusion parameter and $B_t$ is standard Brownian motion. The diffusion model learns the score function $s(X_t, t) = \nabla_{X_t} \log p(X_t)$ using neural network. And the sampling of diffusion process is to solve the probability flow ordinary differential equation:

$$dX_t = (f(X_t, t) - \frac{1}{2}g(t)^2 s(X_t, t))dt, X_T \sim \mathcal{N}(0, I). \tag{2}$$

The two widely adopted diffusion models in text-to-image, namely denoising diffusion probabilistic model (DDPM) and flow matching optimal transport (FM-OT), fit in above framework by setting $f(X_t, t) = -\frac{1}{2}\beta_t X_t, g(t) = \sqrt{\beta_t}$ and $f(X_t, t) = -\frac{1}{1-t}X_t, \frac{1}{2}g(t)^2 = \frac{t}{1-t}$ respectively, where $\beta_t$ is hyper-parameter of DDPM. The forward SDE of DDPM and FM-OT can be directly solved:

$$\text{DDPM: } q(X_t \mid X_0) = \mathcal{N}\big(e^{-\frac{1}{2}\int_0^t \beta_s ds}X_0, \ (1 - e^{-\frac{1}{2}\int_0^t \beta_s ds})I\big), \tag{3}$$

$$\text{FM-OT: } q(X_t \mid X_0) = \mathcal{N}\big((1 - t)X_0, \ t^2 I\big). \tag{4}$$

However, the backward equation in Eq. 2 is intractable as $s(X_t, t)$ is neural network. Usually we need time-consuming multi-step solvers. In this paper, we focus on distilling the solution of backward equations into another neural network.

### 2.2 DISTRIBUTION MATCHING DISTILLATION

From now on we assume a pre-trained diffusion model is available, with learned score function $s_r(X_t, t)$ and distribution $p_r(X_t)$. The Distribution Matching Distillation (DMD) (Yin et al., 2024b;a) distills the diffusion model by a technique named score distillation (Poole et al., 2022). More specifically, DMD learns the generator distribution $p_g(X_t)$ to match the diffusion distribution $p_r(X_t)$:

$$\min_{p_g} D_{KL}(p_g(X_t)||p_r(X_t)) = \mathbb{E}_{t \sim [0,T], p_g}[\log p_g(X_t) - \log p_r(X_t)]. \tag{5}$$

Directly distillation from above target produces suboptimal results. Therefore, DMD introduces an intermediate fake distribution $p_f(X_t, t)$, and optimizes the generator distribution $p_g$ and fake distribution $p_f$ in an interleaved way:

$$\text{Generator: } \min_{p_g} \mathbb{E}_{t \sim [0,T], p_g}[\log p_f(X_t) - \log p_r(X_t)],$$

$$\text{Fake: } \max_{p_f} \mathbb{E}_{t \sim [0,T], p_g}[\log p_f(X_t)]. \tag{6}$$

In practice, the fake distribution is parameterized as the score function $s_\phi(X_t, t) = \nabla \log p_f(X_t)$. On the other hand, the generator is parameterized with a clean image generating network $G_\theta(\epsilon), \epsilon \sim \mathcal{N}(0, I)$ and forward diffusion process $q(X_t|X_0)$, such that $p_g(X_t) = \mathbb{E}_{\epsilon \sim \mathcal{N}(0,I)}[q(X_t|G_\theta(\epsilon))]$. To this end, the DMD updates are achieved by gradient descent and score matching (Vincent, 2011):

$$\text{Generator: } \nabla_\theta \mathcal{L}_g = \mathbb{E}_{t \sim [0,T], \epsilon \sim \mathcal{N}(0,I), X_t \sim q(X_t|G_\theta(\epsilon))}[(s_\phi(X_t, t) - s_r(X_t, t))\frac{\partial X_t}{\partial \theta}],$$

$$\text{Fake: } \nabla_\phi \mathcal{L}_f = \nabla_\phi \mathbb{E}_{t \sim [0,T], \epsilon \sim \mathcal{N}(0,I), X_t \sim q(X_t|G_\theta(\epsilon))}[||s_\phi(X_t, t) - \nabla_{X_t} \log q(X_t|G_\theta(\epsilon))||]. \tag{7}$$

# 3 METHOD: SCALING DISTRIBUTION MATCHING FOR GENERAL DISTILLATION

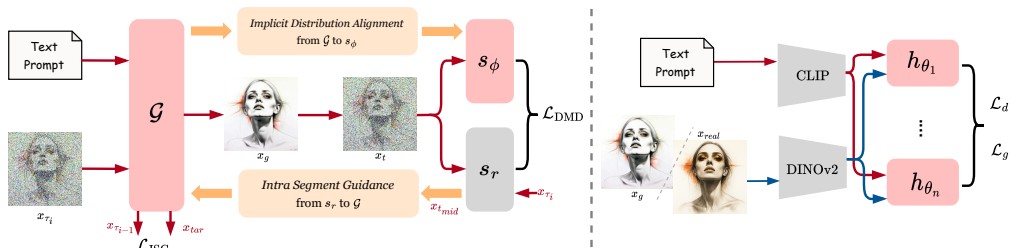

Figure 2: Left: The generator $\mathcal{G}$ receives a text prompt and $x_{\tau_i}$ to produce one-step output $x_g$, which is diffused to $x_t$ and processed by $s_\phi$ and $s_r$ for computing DMD gradient. ISG guides $\mathcal{G}$ using an sampled intermediate $t_{mid}$, and IDA aligns $\mathcal{G}$ with $s_\phi$ after generator update. The overall training pipeline is shown in Algorithm 1. Right: The discriminator extracts semantic features from generated and real images using CLIP and DINOv2, which are processed by head blocks $h_{\theta_i}$ to predict real/fake logits for adversarial training. Trainable modules are shown in pink, while frozen (pretrained) ones are shown in grey.

## 3.1 BOTTLENECKS IN VANILLA DMD: STABILITY, SAMPLING, AND NAIVE DISCRIMINATOR

While Distribution Matching Distillation (DMD) has shown promising results in aligning generative distributions, its vanilla formulation exhibits several fundamental limitations when applied to large-scale models. First, scalability remains a challenge—the two time-scale update rule (TTUR), effective in SD 1.5 (0.8B) and SDXL (2.6B), fails to converge stably when scaled to larger models such as SD 3.5 Large (8B) or FLUX (12B). Second, sampling efficiency is limited as the generator does not incorporate the varying importance of timesteps in the denoising trajectory, which slows convergence and reduces expressiveness. Third, the discriminator lacks generality, with a relatively naive design that struggles to adapt across diverse model scales and architectures. These issues motivate us to propose architectural and algorithmic improvements in this work.

## 3.2 IMPLICIT DISTRIBUTION ALIGNMENT (IDA) FOR FLOW-BASED MODELS

Recall that the DMD can be viewed as a min-max optimization:

$$\min_\theta \max_\phi V(\theta, \phi) \triangleq \mathbb{E}_{t \sim [0,T], p_g}[\log p_f(X_t) - \log p_r(X_t)]. \tag{8}$$

It is obvious that the *inner best response* is attained at $p_f(X_t) = p_g(X_t)$. Furthermore, following Proposition 2 of GAN (Goodfellow et al., 2014), if the inner best response is achieved at every round of generator optimization, then generator converges.

However, the inner best response is brittle. DMD2 uses *two time-scale update rule* (TTUR) (Heusel et al., 2017), which increases the update frequency of fake model. However, at large-scale models such as SD 3.5 Large (8B), simply increasing TTUR ratio is expensive and can still oscillate, as shown in Fig. 3. To tackle the difficulty to achieve inner best response, we introduce a proximal alignment step after each generator update, called *Implicit Distribution Alignment* (IDA):

$$\phi \leftarrow \lambda\phi + (1 - \lambda)\theta, \tag{9}$$

where $\lambda \in (0, 1]$ and close to 1. We claim that IDA maintains $p_f(X_t) \approx p_g(X_t)$, and thus help DMD converges. More formally, we have the following:

**Proposition 3.1.** *Under mild assumptions (Assumptions A.1 and A.6), IDA maintains an $\epsilon$-best inner response. More specifically, after $k$ round of min max*

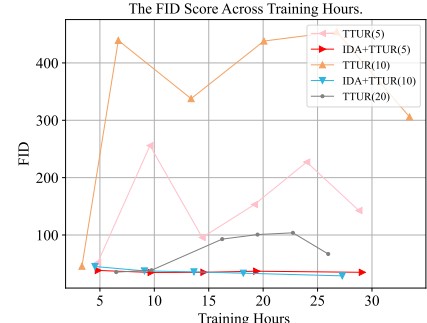

Figure 3: "Training Hours-FID" curves on COCO-5K dataset. When distilling the 8B SD 3.5 Large, IDA improves training stability across TTUR ratios.

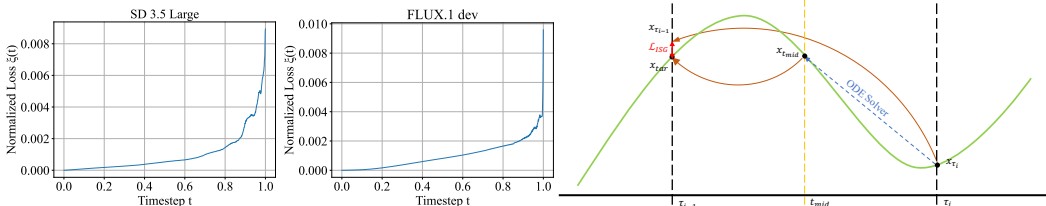

Figure 4: Left: Normalized reconstruction loss $\xi(t)$ over timesteps in $[0, 1]$. Right: Illustration of the Intra-Segment Guidance.

*optimization in Eq. 8, we have*

$$\mathbb{E}_t \, D_{KL}\big(p_g(X_t) \, \| \, p_f(X_t)\big) \; \leq \; \varepsilon, \; i.e., \; p_f(X_t) \approx p_g(X_t) \; (\varepsilon\text{-best response}). \tag{10}$$

In practice, this strategy ensures that the fake distribution remains closely aligned with the generator's distributional trajectory. We observe that combining IDA with even a relatively small TTUR (e.g., 5:1) leads to significantly more stable convergence. An example of this effect is shown in Fig. 3, where we compare FID curves under different TTUR settings with and without IDA. As the figure illustrates, IDA consistently reduces FID variance and improves overall performance.

Intuitively, the generator is trained on a small set of anchor timesteps (e.g., four fixed $t$'s), while the fake model is trained on a much denser grid. The IDA transfers the generator's progress on those anchor $t$'s to the fake model, improving its predictions at those $t$'s. On the other hand, on a SD 3.5 Medium generator, we also observe that DMD training produces generators whose samples are consistent across $4/8/16$ Euler steps, indicating that $G_\theta$ remains close to a smooth continuous-flow integrator. This observation supports using generator parameters to softly anchor the fake model. Proofs, discussion, and visualization results are provided in Appendix §A.

### 3.3 INTRA-SEGMENT GUIDANCE: RELOCATING TIMESTEP IMPORTANCE

The distillation performance of vanilla DMD2 is constrained by supervision at a few handcrafted, coarse timesteps (e.g., $\tau \in \{249, 499, 749, 999\}$). This design has two drawbacks: (i) the generator receives no signal over the rest of the trajectory, hurting generalization; and (ii) the utility of each supervised timestep is highly position-dependent—neighboring timesteps can differ markedly in prediction error. To quantify this local reliability, we visualize the normalized one-step reconstruction loss

$$\xi(t) := \mathbb{E}_{x_0, \, \epsilon \sim \mathcal{N}(0, I)} \Big[ \| \hat{x}_0(x_t, t) - x_0 \|_2^2 \Big], \tag{11}$$

where $x_0$ is sampled from the teacher (SD 3.5 or FLUX.1-dev) and $x_t$ is obtained by the forward diffusion process in Eq. 4. As shown in Fig. 4 (Left), $\xi(t)$ is not monotonic in $t$ and exhibits pronounced local oscillations—especially for $t \in [0.8, 1.0]$. Hence, treating nearby timesteps as equally informative can anchor training to suboptimal points.

To address this, we introduce *Intra-Segment Guidance (ISG)*, which *relocates* the teacher's denoising importance from within each segment $(\tau_{i-1}, \tau_i]$ to its supervised anchor. For every coarse timestep $\tau_i$, we sample an intermediate timestep $t_{\text{mid}} \in (\tau_{i-1}, \tau_i)$. The teacher denoises from $\tau_i$ to $t_{\text{mid}}$ to produce $x_{t_{\text{mid}}}$. The generator then continues from $t_{\text{mid}}$ to $\tau_{i-1}$, yielding the target $x_{\text{tar}}$. In parallel, the generator directly denoises from $\tau_i$ to $\tau_{i-1}$ to produce $x_{\tau_{i-1}}$. We optimize an $\ell_2$ loss that backpropagates only through the generator path:

$$\mathcal{L}_{\text{ISG}}^{(i)} = \mathbb{E}_{\epsilon, \, t_{\text{mid}}} \Big[ \big\| x_{\tau_{i-1}} - \text{stop\_grad}(x_{\text{tar}}) \big\|_2^2 \Big], \tag{12}$$

where $\text{stop\_grad}(\cdot)$ prevents gradients from flowing through the target. By letting each anchor absorb information from its surrounding segment, ISG makes the anchors more representative of local denoising behavior, improving sample quality and training stability.

### 3.4 GENERAL AND POWERFUL DISCRIMINATOR BUILT UPON VISION FOUNDATION MODELS

As shown in Fig. 2, the discriminator $D$ consists of a frozen Vision Foundation Model (VFM) backbone $f_{\text{VFM}}$ with trainable heads $h$, enabling a general yet stable adversarial signal across timesteps.

---

**Algorithm 1** SenseFlow Training Algorithm

---

**Require:** pretrained teacher model $\mu_{\text{real}}$, real dataset $\mathcal{D}_{\text{real}}$, generator update frequency $f$, coarse timestep set $S = \{\tau_0, \tau_1, \tau_2, \tau_3\}$
**Ensure:** trained few-step generator $G$
 1: $G \leftarrow \text{copyWeights}(\mu_{\text{real}})$        ▷ Initialize generator
 2: $\mu_{\text{fake}} \leftarrow \text{copyWeights}(\mu_{\text{real}})$        ▷ Initialize fake distribution network
 3: $D \leftarrow \text{initializeDiscriminator}()$        ▷ Initialize VFM-based discriminator
 4: **for** iteration = 1 to max_iters **do**
 5:      $z \sim \mathcal{N}(0, I)$
 6:      Sample $\tau_i$ from $S$        ▷ Pick timestep for current iteration
 7:      Sample $x_{\text{real}} \sim \mathcal{D}_{\text{real}}$
 8:      **if** $random() < 0.5$ **then**        ▷ With 50% probability, use backward simulation
 9:          $x_{\tau_i} \leftarrow \text{multiStepSampling}(z, \tau_3 \rightarrow \tau_i))$
10:      **else**
11:          $x_{\tau_i} \leftarrow \text{forwardDiffusion}(x_{\text{real}}, \tau_i)$
12:      **end if**
13:      $x \leftarrow G(x_{\tau_i})$
14:      **if** iteration mod $f = 0$ **then**
15:          $\mathcal{L}_{\text{DMD}} \leftarrow \text{distributionMatching}(\mu_{\text{real}}, \mu_{\text{fake}}, x)$
16:          $\mathcal{L}_{\text{G}} \leftarrow -\sigma_{\tau_i}^2 \cdot \mathbb{E}[D(x)]$        ▷ Eq. 15
                                           ▷ Intra-segment guidance (ISG)
17:          $t_{\text{mid}} \sim \mathcal{U}(\tau_i, \tau_{i-1})$
18:          $x_{\text{mid}} \leftarrow \mu_{\text{real}}(x_{\tau_i}, \tau_i \rightarrow t_{\text{mid}})$
19:          $x_{\text{tar}} \leftarrow G(x_{\text{mid}}, t_{\text{mid}} \rightarrow \tau_{i-1})$
20:          $x_{\tau_{i-1}} \leftarrow G(x_{\tau_i}, \tau_i \rightarrow \tau_{i-1})$
21:          $\mathcal{L}_{\text{ISG}} \leftarrow \text{MSE}(x_{\tau_{i-1}}, \text{stopgrad}(x_{\text{tar}}))$
22:          $\mathcal{L}_{\text{G}} \leftarrow \mathcal{L}_{\text{DMD}} + \lambda_{\text{G}} \cdot \mathcal{L}_{\text{G}} + \lambda_{\text{ISG}} \cdot \mathcal{L}_{\text{ISG}}$        ▷ Final loss function for generator
23:          $G \leftarrow \text{update}(G, \mathcal{L}_{\text{G}})$
                                           ▷ Implicit distribution alignment (IDA), as in Eq. 9
24:          $\mu_{fake} \leftarrow \text{IDA}(G, \mu_{fake}, \lambda_{\text{IDA}})$
25:      **end if**
                                           ▷ Update fake score network $\mu_{\text{fake}}$
26:      $t \sim \text{LogitNormalSampling}(0, 1)$        ▷ Using logit-normal density, as in (Esser et al., 2024)
27:      $x_t \leftarrow \text{forwardDiffusion}(\text{stopgrad}(x), t)$
28:      $\mathcal{L}_{\text{denoise}} \leftarrow \text{denoisingLoss}(\mu_{\text{fake}}(x_t, t), \text{stopgrad}(x))$
29:      $\mu_{\text{fake}} \leftarrow \text{update}(\mu_{\text{fake}}, \mathcal{L}_{\text{denoise}})$
                                           ▷ Update discriminator $D$
30:      $\mathcal{L}_{\text{D}} \leftarrow \mathbb{E}[\max(0, 1 - D(x_{\text{real}})] + \mathbb{E}[\max(0, 1 + D(x)]$        ▷ Eq. 14
31:      $D \leftarrow \text{update}(D, \mathcal{L}_{\text{D}})$
32: **end for**

---

Given an input image $x$ and its text prompt, the VFM backbone extracts multi-level semantic features $z = f_{\text{VFM}}(x)$. In addition, we encode the text with CLIP, $c = f_{\text{CLIP}}(\text{text})$, and use VFM features from real images as reference, $r = f_{\text{VFM}}(x^{\text{real}})$, to inject text–image alignment and realism priors. The discriminator is thus

$$D(x, c, r) = h\big(f_{\text{VFM}}(x), c, r\big). \tag{13}$$

These signals allow $D$ to judge both realism and semantic consistency.

**Hinge loss for the discriminator.** We adopt the standard hinge loss:

$$\mathcal{L}_d = \mathbb{E}_{X \sim p_{\text{data}}}[\max(0, 1 - D(X, c, r))] + \mathbb{E}_{\hat{X}_0 \sim p_g}\Big[\max(0, 1 + D(\hat{X}_0, c, r))\Big], \tag{14}$$

where $p_{\text{data}}$ is the empirical distribution of real images and $p_g$ is the generator distribution.

**Adversarial objective for the generator.** During the training process of generator, predictions of $\hat{X}_0$ at large timesteps can be less reliable. To avoid overpowering the DMD signal under high noise, we introduce a weighting mechanism. Specifically, we scale the adversarial term by the *signal*

*power* of the current timestep. Let the forward process be $x_t = \alpha_t x_0 + \sigma_t \epsilon$ (see Sec. 2). We define $\omega(t) = (1 - \sigma_t)^2$, which decreases as noise increases, and optimize

$$\mathcal{L}_g = -\omega(t) \cdot \mathbb{E}_{\hat{X}_0 \sim p_g} \left[ D(\hat{X}_0, c, r) \right] = -\alpha_t^2 \cdot \mathbb{E}_{\hat{X}_0 \sim p_g} \left[ D(\hat{X}_0, c, r) \right]. \qquad (15)$$

This weighting emphasizes the DMD gradient at noisy, high-$t$ steps while leveraging GAN feedback more strongly at cleaner, low-noise steps, improving stability and overall quality in practice. Implementation details of the discriminator are provided in Appendix B.2.

**Overall training pipeline.** Algorithm 1 summarizes the complete SenseFlow optimization loop. At each iteration, we sample a coarse timestep $\tau_i \in S$ and construct $x_{\tau_i}$ either by backward simulation from noise or by forward diffusion from real data. We update the generator every $f$ iterations (TTUR-style) with the combined objective of DMD, the VFM-based adversarial term, and ISG. After each generator update, we apply IDA to softly anchor the fake model toward the updated generator. Finally, we update the fake model and the discriminator with their respective losses.

# 4 EXPERIMENTAL RESULTS

## 4.1 EXPERIMENTAL SETUP

**Datasets and Benchmarks.** Following DMD2 (Yin et al., 2024a), our experiments are conducted using a filtered set of the LAION-5B (Schuhmann et al., 2022) dataset, which provides high-quality image-text pairs for training. We select images with a minimum aesthetic score (aes score) of 5.0 and a shorter dimension of at least 1024 pixels, ensuring the dataset comprises visually appealing, high-resolution images suitable for our model's requirements.

For evaluation, we construct a validation set using the COCO 2017 (Lin et al., 2014) validation set, which contains 5,000 images. Each image in this set is paired with the text annotation that yields the highest CLIP Score (ViT-B/32), thus forming a robust text-image validation set. To assess compositional generalization, we further evaluate on GenEval (Ghosh et al., 2023)—which programmatically checks object presence, attributes, relations, and counting-and T2I-CompBench (Huang et al., 2023)—which covers attribute binding, inter-object relations, and complex multi-object compositions; we follow the official protocols and report both overall and per-category scores.

**Text-to-Image Models**. We conduct extensive experiments on three representative large-scale text-to-image models: FLUX.1 dev (12B) (Labs, 2024), Stable Diffusion 3.5 Large (8B) (Esser et al., 2024), and SDXL (2.6B) (Podell et al., 2024), which span different model sizes and generative paradigms. Results demonstrate the generality and effectiveness of our method across both flow-based and conventional diffusion architectures.

**Evaluation Metrics.** Following (Wang et al., 2024a; Lin et al., 2024; Yin et al., 2024a), we report FID and Patch FID of all baselines and the generated images of original teacher models to assess distillation performance and high-resolution details, dubbed FID-T and Patch FID-T. For semantic faithfulness, image quality and human preference, we additionally report CLIP Score (ViT-B/32) (Radford et al., 2021), HPS v2 (Wu et al., 2023) (a human-preference predictor), ImageReward (Xu et al., 2023) (a learned reward approximating human judgments), and PickScore (Kirstain et al., 2023) (trained on pairwise human choices), which complement FID by focusing on perceived quality and semantic alignment.

## 4.2 TEXT TO IMAGE GENERATION

**Comparison Baselines.** For *SDXL* distillation, we compare against LCM (Luo et al., 2023a), PCM (Wang et al., 2024a), Flash Diffusion (Chadebec et al.), SDXL-Lightning (Lin et al., 2024), Hyper-SD (Ren et al., 2024), and DMD2 (Yin et al., 2024a). For *SD 3.5 Large*, we use SD 3.5 Large Turbo as the best baseline (Sauer et al., 2024b). For *FLUX.1 dev*, we compare with Hyper-FLUX (Ren et al., 2024), FLUX.1 schnell (Labs, 2024), and FLUX-Turbo-Alpha (Team, 2024). All baselines are evaluated under their official 4-step configurations.

**Quantitative Comparison.** We evaluate 4-step models on COCO-5K and GenEval (Table 1) and report T2I-CompBench results (Table 5). For flow-matching models (SD 3.5 Large and FLUX.1 dev) we include both stochastic and deterministic solvers, denoted as "Ours" and "Ours (Euler)".

Table 1: Quantitative Results on COCO-5K and GenEval Benchmarks. **Bold**/Underline: best/second best in distilling the same teacher. Our method achieves superior performance on 4-step generation.

| Method | #NFE↓ | Patch FID-T↓ | CLIP↑ | HPSv2↑ | Pick↑ | IR↑ | GenEval↑ |
|---|---|---|---|---|---|---|---|
| **Stable Diffusion XL Comparison** | | | | | | | |
| SDXL (Podell et al., 2024) | 80 | – | 0.3293 | 0.2930 | 22.67 | 0.8719 | 0.5461 |
| LCM-SDXL (Luo et al., 2023a) | 4 | 30.63 | 0.3230 | 0.2824 | 22.22 | 0.5693 | 0.5036 |
| PCM-SDXL (Wang et al., 2024a) | 4 | 17.77 | 0.3242 | 0.2920 | 22.54 | 0.6926 | 0.4944 |
| Flash-SDXL (Chadebec et al.) | 4 | 23.24 | 0.3216 | 0.2830 | 22.17 | 0.4295 | 0.4715 |
| SDXL-Lightning (Lin et al., 2024) | 4 | **16.57** | 0.3214 | 0.2931 | 22.80 | 0.7799 | 0.5332 |
| Hyper-SDXL (Ren et al., 2024) | 4 | 17.49 | 0.3254 | 0.3000 | 22.98 | 0.9777 | 0.5398 |
| DMD2-SDXL (Yin et al., 2024a) | 4 | 18.72 | **0.3277** | 0.2963 | 22.98 | 0.9324 | 0.5779 |
| Ours-SDXL | 4 | 21.01 | 0.3248 | **0.3010** | **23.17** | **0.9951** | **0.5784** |
| **Stable Diffusion 3.5 Large Comparison** | | | | | | | |
| SD 3.5 (Esser et al., 2024) | 80 | – | 0.3310 | 0.2993 | 22.98 | 1.1629 | 0.7140 |
| SD 3.5 Turbo (Sauer et al., 2024b) | 4 | 22.88 | 0.3262 | 0.2909 | 22.89 | 1.0116 | 0.6877 |
| Ours-SD 3.5 | 4 | **17.48** | 0.3286 | **0.3016** | **23.01** | 1.1713 | 0.6955 |
| Ours-SD 3.5 (Euler) | 4 | 20.26 | **0.3287** | 0.3008 | 22.90 | **1.2062** | **0.7098** |
| **FLUX Comparison** | | | | | | | |
| FLUX.1 dev (Labs, 2024) | 50 | – | 0.3202 | 0.3000 | 23.18 | 1.1170 | 0.6699 |
| | 25 | – | 0.3207 | 0.2986 | 23.14 | 1.1063 | 0.6733 |
| FLUX.1-schnell (Labs, 2024) | 4 | – | 0.3264 | 0.2962 | 22.77 | 1.0755 | 0.6807 |
| Hyper-FLUX (Ren et al., 2024) | 4 | 23.47 | **0.3238** | 0.2963 | 23.09 | 1.0983 | 0.6193 |
| FLUX-Turbo-Alpha (Team, 2024) | 4 | 24.52 | 0.3218 | 0.2907 | 22.89 | 1.0106 | 0.4724 |
| Ours-FLUX | 4 | **19.60** | 0.3167 | 0.2997 | 23.13 | 1.0921 | **0.6471** |
| Ours-FLUX (Euler) | 4 | 20.29 | 0.3171 | **0.3008** | **23.26** | **1.1424** | 0.6420 |

On **COCO-5K & GenEval,** across all teachers, our 4-step distillation performs strongly on modern, human-correlated metrics. On SD 3.5, "Ours-SD 3.5" and "Ours-SD 3.5 (Euler)" achieve the best and second-best scores *on all metrics*, even surpassing the teacher model in HPSv2, PickScore, and ImageReward. The Euler variant achieves the highest GenEval (0.7098), approaching the 80-NFE teacher (0.7140). On SDXL, our distilled model ranks first on most metrics, including HPSv2, PickScore, ImageReward and GenEval, with CLIP close to prior art and competitive Patch FID-T. On FLUX.1 dev, our models again deliver best and second-best performance across five of six metrics. The Euler variant further surpasses the teacher model in HPSv2, PickScore, and ImageReward. As for **T2I-CompBench,** our SD 3.5 (Euler) model is best in five of six categories and second on "Spatial" category, establishing overall *state-of-the-art* performance. For SDXL, our model is best or second-best in all six categories, giving the *strongest* overall SDXL distillation on this benchmark. For FLUX, the Euler variant is best or second in three of six categories, achieving the overall *second best* performance. Detailed results of GenEval and T2I-CompBench are shown in Appendix B.4.

Overall results indicate that our method preserves fidelity while improving semantic alignment and compositional correctness, as reflected by GenEval and T2I-CompBench across diverse models.

**Qualitative Comparison.** Fig 5 presents qualitative comparisons across a set of prompts. Our method generates images with sharper details, better limb structure, and more coherent lighting dynamics, compared to teacher models and baselines. Notably, "Ours-SD3.5" and "Ours-FLUX" produce more faithful and photorealistic generations under challenging prompts involving fine textures, human faces, and scene lighting. Additional visualization results are provided in the appendix.

## 4.3 ABLATION STUDIES

**Effectiveness of Implicit Distribution Alignment.** To assess the effectiveness of our proposed IDA, we conduct experiments on SD 3.5 Large with various TTUR ratios. As shown in Fig. 3, we compare FID curves across different settings, both with and without IDA. Without IDA, the curves corresponding to "TTUR(5)", "TTUR(10)", and "TTUR(20)" exhibit severe oscillations, indicating unstable training dynamics and unreliable optimization of the fake distribution—even at a high ratio of 20:1. This instability leads to inaccurate DMD gradients and poor convergence. In contrast, the

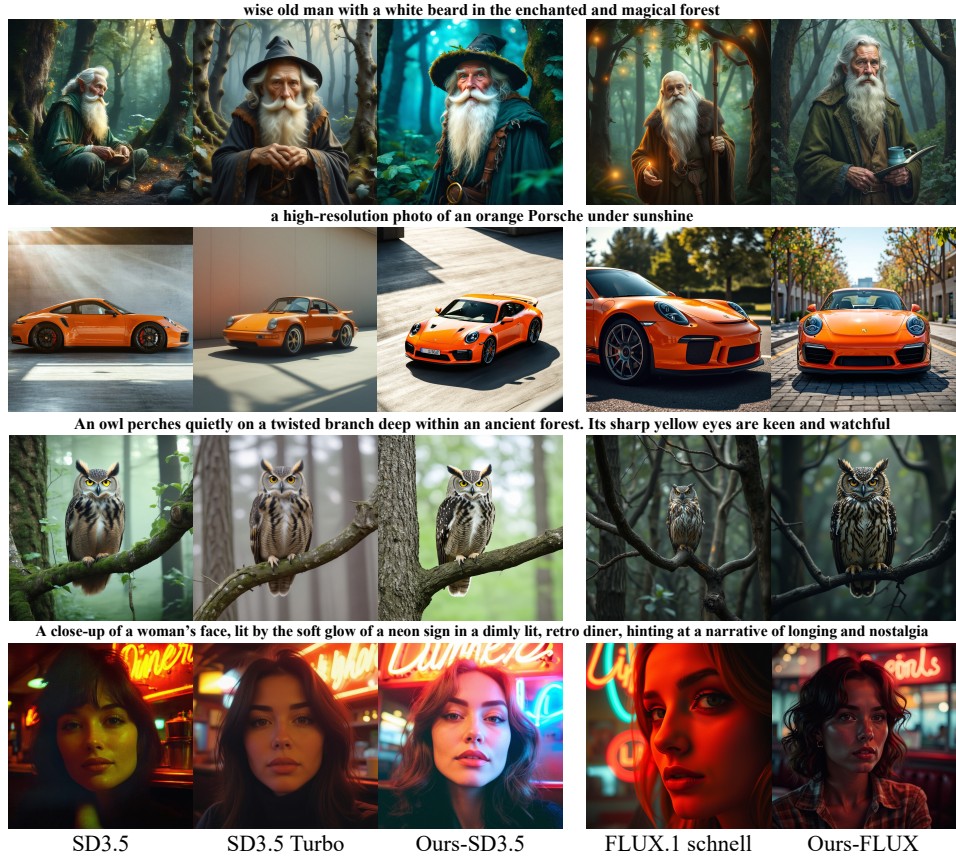

**wise old man with a white beard in the enchanted and magical forest**

**a high-resolution photo of an orange Porsche under sunshine**

**An owl perches quietly on a twisted branch deep within an ancient forest. Its sharp yellow eyes are keen and watchful**

**A close-up of a woman's face, lit by the soft glow of a neon sign in a dimly lit, retro diner, hinting at a narrative of longing and nostalgia**

|  SD3.5 | SD3.5 Turbo | Ours-SD3.5 | FLUX.1 schnell | Ours-FLUX |

Figure 5: Qualitative comparisons on challenging prompts across methods. Our method shows superior fidelity, especially in rendering human faces, scene composition, and fine-grained textures.

settings that incorporate IDA (i.e., "IDA+TTUR(5)" and "IDA+TTUR(10)") demonstrate significantly smoother and more stable FID reductions, highlighting IDA's ability to stabilize training and improve convergence, even at a relatively small TTUR ratio (5:1).

In addition to the FID analysis, we report quantitative comparisons in Tab. 2 between "w/o ISG" and "w/o ISG, w/o IDA" using five metrics: FID-T, HPSv2, PickScore, ImageReward, and AESv2. Across all metrics, adding IDA leads to consistent improvements, further confirming that IDA plays a key role in enhancing training stability and distillation quality. More observation results in Appendix A.5 also serve as evidences for IDA as the proposition of $\varepsilon$-best inner response.

Table 2: Ablation Study Results of IDA, ISG, and VFM Discriminator.

| Method | FID-T↓ | HPSv2↑ | Pick↑ | IR↑ | AESv2↑ |
|---|---|---|---|---|---|
| **Stable Diffusion 3.5 Large** | | | | | |
| Ours | **13.38** | **0.3015** | **23.03** | **1.1713** | **5.482** |
| w/o ISG | 17.00 | 0.2971 | 22.75 | 1.0186 | 5.453 |
| w/o ISG, w/o IDA | 43.84 | 0.2555 | 20.60 | 0.3828 | 5.102 |
| **Stable Diffusion XL** | | | | | |
| DMD2-SDXL | **15.04** | 0.2964 | 22.98 | 0.9324 | 5.530 |
| DMD2 w VFM | 18.55 | **0.2995** | **23.00** | **0.9744** | **5.625** |

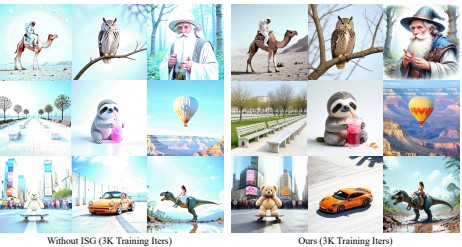

Without ISG (3K Training Iters)  Ours (3K Training Iters)

Figure 6: The ISG improves training consistency, especially in early stage of training.

**Effectiveness of Intra-Segment Guidance.** To evaluate the effectiveness of the Intra-Segment Guidance (ISG) module during distillation, we conduct an ablation study on Stable Diffusion 3.5 Large. As shown in Tab. 2, we compare our model with and without ISG (denoted as "Ours" and "w/o ISG", respectively) on the COCO-5K dataset. The results indicate that incorporating ISG leads

to significant improvements across all aspects, including image quality, text-image alignment, and human preference quality.

Fig. 6 shows a qualitative snapshot at 3k training iterations (generator updated for 300 steps with a 10:1 TTUR). With ISG, generations are visibly more consistent and semantically accurate, while the model without ISG exhibits color shifts and degraded details. This supports our interpretation of ISG as a segment-aware supervision that stabilizes early optimization and accelerates convergence.

**Training-time overhead.** We evaluate the running time of our introduced IDA and ISG. In our TTUR setting ($f$=5), ISG and IDA run only on *generator* steps, amortizing their cost. Across three backbones, enabling **ISG** increases per-iteration time by **+4.44%** (SDXL), **+3.23%** (FLUX), and **+6.16%** (SD 3.5). Adding **IDA** on top of an ISG-free baseline adds only **+3.97%** (FLUX) and **+0.57%** (SD 3.5). These overheads are modest compared to the gains in convergence stability and sample quality; full timing statistics are in Appendix. B.3, Tab. 3.

**VFM-based Discriminator.** We ablate the discriminator on the SDXL backbone by replacing the diffusion-based discriminator in DMD2-SDXL with our VFM-based design ("DMD2 w VFM"). As reported in Tab. 2, the VFM discriminator improves human-centric metrics and aesthetics—HPSv2, PickScore, ImageReward, and AESv2—while showing a trade-off on FID-T. These results suggest that leveraging VFM features provides stronger semantic and stylistic priors for adversarial feedback, demonstrating better generalization and alignment with human preferences.

**FID-T vs. human-preference trade-off.** Adding the VFM discriminator raises FID-T, as shown in Tab. 2. However, We suggest that human-preference metrics (HPSv2, PickScore, etc) mainly capture perceived quality and semantic faithfulness, whereas FID-T is sensitive to both quality *and* diversity relative to the teacher. The VFM discriminator imposes a semantic prior, nudging the generator toward VFM-preferred modes: this improves alignment with human-preferred semantics but may reduce sample variance of less-favored regions, hence the modest rise in FID-T. In few-step distillation context, we prioritize high-quality, semantically meaningful modes over exhaustive distribution coverage; We view this quality–diversity trade-off acceptable in practical use.

## 5 RELATED WORK

**Diffusion Distillation** methods mainly fall into two categories: trajectory-based and distribution-based approaches. Trajectory-based methods, such as Direct Distillation (Luhman & Luhman, 2021) and Progressive Distillation (Salimans & Ho, 2022; Ren et al., 2024; Lin et al., 2024; Chadebec et al.), learn to replicate the denoising trajectory, while Consistency Models (Song et al., 2023; Luo et al., 2023a; Kim et al., 2024; Wang et al., 2024a; Lu & Song, 2024; Chen et al., 2025) enforce self-consistency across steps. Distribution-based methods aim to match the generative distribution, including GAN-based distillation (Wang et al., 2023a; 2024b; Luo et al., 2024) and VSD variants (Wang et al., 2023b; Yin et al., 2024b;a). ADD (Sauer et al., 2024b) and LADD (Sauer et al., 2024a) explored distilling diffusion models using adversarial training with pretrained feature extractors. RayFlow (Shao et al., 2025) explored sampling important timesteps for better distillation. Among these, DMD2 (Yin et al., 2024a) has shown strong results on standard diffusion models (e.g., SDXL), but its stability degrades on large-scale models. Our work builds upon DMD2 and addresses these limitations by introducing SenseFlow, which scales distribution matching distillation to SD 3.5 and FLUX.1 dev through improved alignment and regularization strategies.

## 6 DISCUSSION & CONCLUSION

We scale up distribution-matching distillation for large flow-based models via *implicit distribution alignment* and *intra-segment guidance*; together with a VFM-based discriminator, these yield our *SenseFlow*, which achieves stable and effective few-step generation across both diffusion and flow-matching backbones. Across three teachers—SDXL, SD 3.5 Large, and FLUX.1 dev—SenseFlow attains superior overall 4-step results on modern human-preference and compositional benchmarks. Looking ahead, we aim to push to more aggressive sampling regimes (2-step and 1-step) and to study alternative vision backbones for the discriminator/guidance modules (Oquab et al., 2023; Ravi et al., 2025; Ranzinger et al., 2024; He et al., 2022).

ACKNOWLEDGEMENT

This work was supported by the Hong Kong Research Grants Council under the Areas of Excellence scheme grant AoE/E-601/22-R and NSFC/RGC Collaborative Research Scheme grant CRS_HKUST603/22.

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

## A  THEORY PROOFS, DISCUSSION, AND ADDITIONAL RESULTS FOR IDA

**Symbol Definitions.**  We provide explicit symbol definitions and clarifications for our IDA method, which is primarily applied to flow matching-based text-to-image models such as SD 3.5 Medium/Large and FLUX.1 dev. Since these models operate within the flow matching framework, we formalize the relevant notation used throughout the analysis and proof.

Let the generator be $G_\theta$, which maps a random noise $\epsilon \sim \mathcal{N}(0, I)$ to a generated sample $G_\theta(\epsilon)$. The fake/teacher models predict velocity fields $v_\phi(x, t)$ and $v_r(x, t)$, corresponding to the fake model and the teacher model, respectively.

Given the forward kernel $q(X_t \mid X_0)$ as described in Eq. 4, the generator induces $t$-marginals:

$$p_g(X_t) = \mathbb{E}_{\epsilon \sim \mathcal{N}(0, I)}[q(X_t \mid G_\theta(\epsilon))], \tag{16}$$

where $p_g(X_t)$ is the distribution induced by the generator samples at timestep $t$. Similarly, the teacher and fake models define $p_r(X_t)$ and $p_f(X_t)$, respectively.

**Generator-induced FM target field and tracking objective (inner loop of Eq. 8).**  Let $(X_0, X_t, X_1)$ denote the random FM path induced by the forward kernel $q(X_t \mid X_0)$, where $X_0 \sim p_g$ and $X_1 \sim \mathcal{N}(0, I)$ is the Gaussian noise endpoint. We define the generator-induced FM target field as

$$\hat{v}_\theta(x, t) := \mathbb{E}[X_1 - X_0 \mid X_t = x]. \tag{17}$$

Using this target, we define the *tracking error* at the $k$-th round of min–max optimization:

$$\bar{d}_k = \mathbb{E}_{t, X_t} \| v_{\phi_k}(X_t, t) - \hat{v}_{\theta_k}(X_t, t) \|. \tag{18}$$

This is the quantity directly optimized by the inner loop: a smaller $\bar{d}_k$ indicates that the fake model (and thus $p_f$) better tracks the generator-induced path distribution $p_g$.

In our setup, the generator shares the same network architecture as the teacher model and is also implemented as a time-conditional velocity network, with output defined for any $t \in [0, 1]$. Here, we define the *generator-parameterized* velocity field $v_\theta(x, t)$ as the output of the generator's network at timestep $t$:

$$v_\theta(x, t) := f_\theta(x, t), \qquad t \in [0, 1], \tag{19}$$

*Notation.* We abuse notation slightly and write $v_\theta(X_t, t)$ to denote evaluating the deterministic field $v_\theta(\cdot, \cdot)$ at the random input $(X_t, t)$. In addition, during DMD training, supervision is applied only at a sparse set of anchor timesteps for the generator, making it a few-step generator that can produce clean samples in a small number of steps (e.g., 4). However, this sparsity does not restrict the definition of $v_\theta(x, t)$ to the anchors: since the generator is time-conditional, we can still evaluate its output for any $t \in [0, 1]$; anchors only specify where gradients are applied during training.

**Self-consistent error.**  We further define the *self-consistent error* at round $k$ as

$$\bar{\beta}_k = \mathbb{E}_{t, X_t} \| v_{\theta_k}(X_t, t) - \hat{v}_{\theta_k}(X_t, t) \|. \tag{20}$$

The self-consistent error quantifies how well the generator's own velocity prediction agrees with its pathwise FM target along the generator sample-induced trajectory. Our analysis assumes $\sup_k \bar{\beta}_k < \infty$, which is supported empirically by step consistency results in Appendix A.5.

**Assumption A.1** (Local regularity). There exists a neighborhood $\mathcal{U} \subset \mathbb{R}^d$ of the parameter trajectory and constants $L, C_v, C_{\hat{v}} > 0$ such that:

**(Param→field Lipschitz)**  $\mathbb{E}_{t, X_t \sim p_g} \| v_{\omega'}(X_t, t) - v_\omega(X_t, t) \| \le L \| \omega' - \omega \|, \qquad \forall \omega, \omega' \in \mathcal{U},$ $\tag{21}$

**(Generator smoothness)**  $\mathbb{E}_{t, X_t \sim p_g} \| v_{\theta_{k+1}}(X_t, t) - v_{\theta_k}(X_t, t) \| \le C_v \| \theta_{k+1} - \theta_k \|,$

**(Target smoothness)**  $\mathbb{E}_{t, X_t \sim p_g} \| \hat{v}_{\theta_{k+1}}(X_t, t) - \hat{v}_{\theta_k}(X_t, t) \| \le C_{\hat{v}} \| \theta_{k+1} - \theta_k \|.$

In particular, the param→field Lipschitz condition applies to both the fake and generator parameters, e.g., to the pair $(\omega', \omega) = (\phi_{k+1}, \theta_k)$. We also assume the min–max optimization steps satisfy $\| \theta_{k+1} - \theta_k \| \to 0$, since the generator will gradually converge as the training process.

## A.1 DECOMPOSITION AND INNER BEST RESPONSE

**Proposition A.2** (Cross-entropy decomposition and best response). *With $V(\theta, \phi)$ defined in Eq. 8, for fixed $\theta$,*

$$V(\theta, \phi) = \mathbb{E}_t \, D_{\mathrm{KL}}\big(p_g(X_t) \,\|\, p_r(X_t)\big) - \mathbb{E}_t \, D_{\mathrm{KL}}\big(p_g(X_t) \,\|\, p_f(X_t)\big). \tag{22}$$

*Hence $\max_\phi V(\theta, \phi) = \mathbb{E}_t \, D_{\mathrm{KL}}(p_g \| p_r)$, attained if $p_f(X_t) = p_g(X_t)$ for a.e. t.*

*Proof.* Use the cross-entropy identity $\mathbb{E}_{p_g}[\log p_f] = -D_{\mathrm{KL}}(p_g \| p_f) - H(p_g)$ and the fact that $H(p_g)$ cancels. Non-negativity of KL yields the claim. □

## A.2 BOUNDING THE TRACKING ERROR UNDER IDA UPDATE

Firstly, we define the parameter tracking error $e_k := \|\phi_k - \theta_k\|$, and the field gap $\Delta_k := \mathbb{E}_{t, X_t} \|v_{\phi_k}(X_t, t) - v_{\theta_k}(X_t, t)\|$. Meanwhile, as defined in Eq. 18 and Eq. 20, we have the target field tracking and self-consistent errors $\bar{d}_k$ and $\bar{\beta}_k$. In this subsection, we use the IDA update $\phi_{k+1} = \lambda \phi_k + (1 - \lambda)\theta_{k+1}$ (Eq. 9) to control the parameter tracking error and the field gap.

*What we prove.* With the IDA update, we (i) bound the next-step field gap in terms of the parameter tracking error $e_k$ and the outer step $\|\theta_{k+1} - \theta_k\|$, (ii) derive a coupled one-step recursion for $(e_k, \bar{d}_k)$, and (iii) obtain the asymptotic bound $\limsup_{k \to \infty} \bar{d}_k \le \bar{\beta}_\star$.

**Lemma A.3** (Field-gap bound under IDA using $e_k$). *Under Assumption A.1,*

$$\Delta_{k+1} \le L\lambda \, e_k + \big[L(1 - \lambda) + C_v\big] \|\theta_{k+1} - \theta_k\|. \tag{23}$$

*Proof.* Insert and subtract $v_{\theta_k}$ and apply generator smoothness and the param$\to$field Lipschitz:

$$\Delta_{k+1} \le \mathbb{E}\|v_{\phi_{k+1}} - v_{\theta_k}\| + \mathbb{E}\|v_{\theta_k} - v_{\theta_{k+1}}\|$$
$$\le L\|\phi_{k+1} - \theta_k\| + C_v\|\theta_{k+1} - \theta_k\|.$$

Using the IDA update, $\phi_{k+1} - \theta_k = \lambda(\phi_k - \theta_k) + (1 - \lambda)(\theta_{k+1} - \theta_k)$, so by the triangle inequality

$$\|\phi_{k+1} - \theta_k\| \le \lambda\|\phi_k - \theta_k\| + (1 - \lambda)\|\theta_{k+1} - \theta_k\| = \lambda e_k + (1 - \lambda)\|\theta_{k+1} - \theta_k\|.$$

Plug this into the previous display to obtain Eq. 23. □

**Lemma A.4** (Coupled one-step recursion for tracking). *Under Assumption A.1 and Eq. 9, the following coupled recursions hold:*

$$e_{k+1} \le \lambda \, e_k + \lambda \|\theta_{k+1} - \theta_k\|, \tag{24}$$

$$\bar{d}_{k+1} \le L\lambda e_k + \bar{\beta}_k + \underbrace{\big[L(1 - \lambda) + 2C_v + C_{\hat{v}}\big]}_{=:K} \|\theta_{k+1} - \theta_k\|. \tag{25}$$

*Proof.* For Eq. 24, from $\phi_{k+1} = \lambda \phi_k + (1 - \lambda)\theta_{k+1}$ we have $e_{k+1} = \|\lambda(\phi_k - \theta_{k+1})\| \le \lambda e_k + \lambda\|\theta_{k+1} - \theta_k\|$.

For Eq. 25, by the triangle inequality,

$$\bar{d}_{k+1} = \mathbb{E}\|v_{\phi_{k+1}} - \hat{v}_{\theta_{k+1}}\| \le \underbrace{\mathbb{E}\|v_{\phi_{k+1}} - v_{\theta_{k+1}}\|}_{\Delta_{k+1}} + \underbrace{\mathbb{E}\|v_{\theta_{k+1}} - \hat{v}_{\theta_{k+1}}\|}_{\bar{\beta}_{k+1}}.$$

Apply Lemma A.3 to $\Delta_{k+1}$ and the generator/target smoothness to $\bar{\beta}_{k+1}$:

$$\bar{\beta}_{k+1} \le \bar{\beta}_k + (C_v + C_{\hat{v}})\|\theta_{k+1} - \theta_k\|.$$

Combining the bounds and using $\Delta_{k+1} \le L\lambda e_k + [L(1 - \lambda) + C_v]\|\theta_{k+1} - \theta_k\|$ gives Eq. 25. □

**Proposition A.5** (Asymptotic bound for the tracking error). *If $\sup_k \bar{\beta}_k \le \bar{\beta}_\star$ for some finite $\bar{\beta}_\star < \infty$ and $\|\theta_{k+1} - \theta_k\| \to 0$, then*

$$\limsup_{k \to \infty} \bar{d}_k \le \bar{\beta}_\star.$$

*Proof.* From Eq. 24, $(1 - \lambda)\limsup e_k \le \lambda \limsup \|\theta_{k+1} - \theta_k\| = 0$, so $\limsup e_k = 0$. Taking $\limsup$ in Eq. 25 and using $\limsup \|\theta_{k+1} - \theta_k\| = 0$ together with $\limsup \bar{\beta}_k \le \bar{\beta}_\star$ yields the claim. □

## A.3 From velocity field error to KL gap along the FM path

In this subsection, given pairs of velocity fields $(v_{\phi_k}, v_{\theta_k})$ at round $k$, we relate their discrepancy to the pathwise KL divergence between $p_g(X_t)$ and $p_f(X_t)$.

**Assumption A.6** (Fisher→KL control and score–velocity relation). For each $t \in [0, 1]$, there exists a constant $C_t > 0$ such that

$$D_{\mathrm{KL}}\big(p_g(X_t) \,\|\, p_f(X_t)\big) \;\leq\; C_t \, \mathbb{E}_{X_t \sim p_g} \| \, s_f(X_t, t) - s_g(X_t, t) \, \|_2^2, \tag{26}$$

where $s_\bullet(\cdot, t) = \nabla_x \log p_\bullet(\cdot, t)$ denotes the score. Moreover, along the (linear–Gaussian) flow-matching path, the score and velocity differences are related by a scalar factor $a(t) > 0$ that depends only on the time schedule:

$$s_f(X_t, t) - s_g(X_t, t) \;=\; a(t) \big( v_\phi(X_t, t) - v_\theta(X_t, t) \big). \tag{27}$$

Assume $t \mapsto C_t$ and $t \mapsto a(t)$ are measurable and that

$$C \;:=\; \mathbb{E}_t\big[C_t \, a(t)^2\big] \;<\; \infty. \tag{28}$$

The constants $C_t$ and $a(t)$ are independent of $(\phi, \theta)$.

*Immediate consequence.* Combining Eq. 26–Eq. 27 and averaging over $t$ yields

$$\mathbb{E}_t \, D_{\mathrm{KL}}\big(p_g(X_t) \,\|\, p_f(X_t)\big) \;\leq\; C \, \mathbb{E}\| \, v_\phi(X_t, t) - v_\theta(X_t, t) \, \|_2^2. \tag{29}$$

**Proposition A.7** (From field error to $\varepsilon$-best response). *Under Assumptions A.1 and A.6, define the $L_2$ versions of the tracking and self-consistent errors*

$$\tilde{d}_k \;:=\; \big(\mathbb{E}\|v_{\phi_k}(X_t, t) - \hat{v}_{\theta_k}(X_t, t)\|_2^2\big)^{1/2}, \qquad \tilde{\beta}_k \;:=\; \big(\mathbb{E}\|v_{\theta_k}(X_t, t) - \hat{v}_{\theta_k}(X_t, t)\|_2^2\big)^{1/2}.$$

*Then for each iterate $k$ there exists*

$$\varepsilon_k \;=\; 2\,C\big(\tilde{d}_k^2 + \tilde{\beta}_k^2\big)$$

*such that*

$$\mathbb{E}_t \, D_{\mathrm{KL}}\big(p_g(X_t) \,\|\, p_f(X_t)\big) \;\leq\; \varepsilon_k.$$

*It controls the inner KL gap, and thus induces an $\varepsilon_k$-best-response bound.*

*Proof.* By Assumption A.6, $\mathbb{E}_t D_{\mathrm{KL}}(p_g\|p_f) \leq C \, \mathbb{E}\|v_\phi - v_\theta\|_2^2$. Decompose $v_\phi - v_\theta = (v_\phi - \hat{v}_\theta) + (v_\theta - \hat{v}_\theta)$ and apply $\|u + v\|_2^2 \leq 2\|u\|_2^2 + 2\|v\|_2^2$:

$$\mathbb{E}\|v_\phi - v_\theta\|_2^2 \;\leq\; 2\,\mathbb{E}\|v_\phi - \hat{v}_\theta\|_2^2 + 2\,\mathbb{E}\|v_\theta - \hat{v}_\theta\|_2^2 \;=\; 2(\tilde{d}_k^2 + \tilde{\beta}_k^2).$$

Combine with A.6 to obtain the claim. $\qquad\square$

## A.4 Combining them together

Combining the IDA-specific bound on the field tracking error (Proposition A.5) with the KL control in Proposition A.7, we obtain an $\varepsilon$-best-response guarantee for the inner loop under IDA. Finally, invoking the decomposition in Proposition A.2, we have

$$V(\theta, \phi) = \mathbb{E}_t D_{\mathrm{KL}}(p_g \,\|\, p_r) - \mathbb{E}_t D_{\mathrm{KL}}(p_g \,\|\, p_f).$$

With $\mathbb{E}_t \, D_{\mathrm{KL}}(p_g\|p_f) \leq \varepsilon$ from Proposition A.7, we obtain

$$\mathbb{E}_t \, D_{\mathrm{KL}}(p_g\|p_r) - \varepsilon \;\leq\; V(\theta, \phi_{\mathrm{IDA}}) \;\leq\; \mathbb{E}_t \, D_{\mathrm{KL}}(p_g\|p_r). \tag{30}$$

**In words, under IDA the inner loop delivers an $\varepsilon$-best response, so the outer loop approximately minimizes $\mathbb{E}_t D_{\mathrm{KL}}(p_g\|p_r)$ within $\varepsilon$.**

## A.5 Observations of step consistency and empirical boundedness of the self-consistency error

This subsection provides empirical evidence that the student generator exhibits *step consistency* (similar outputs under 4/8/16 FM integration steps), which in turn indicates that the *self-consistent error* $\bar{\beta}_k = \mathbb{E}_{t, X_t}\|v_{\theta_k}(X_t, t) - \hat{v}_{\theta_k}(X_t, t)\|$ remains bounded during training—supporting the $\varepsilon$-best-response analysis in §3.2 and A.4.

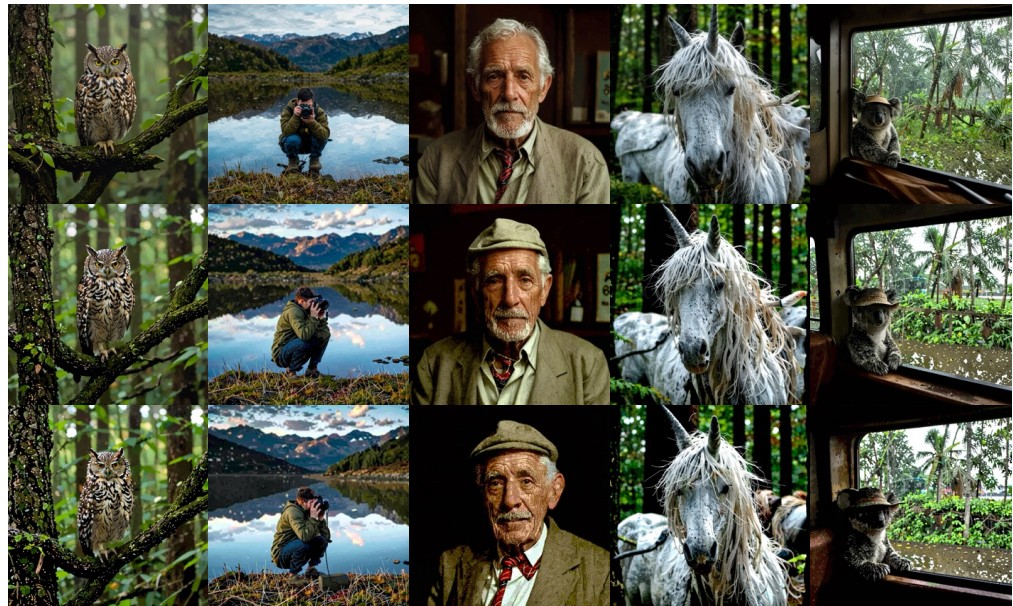

Figure 7: Step consistency (DMD + VFM, TTUR 5:1). Each column shares prompt/seed; rows use 4/8/16 Euler steps. Despite being trained on four anchor timesteps, the 4-step generator remains visually close when re-evaluated under 8 or 16 steps, suggesting a smooth underlying velocity field along the FM path.

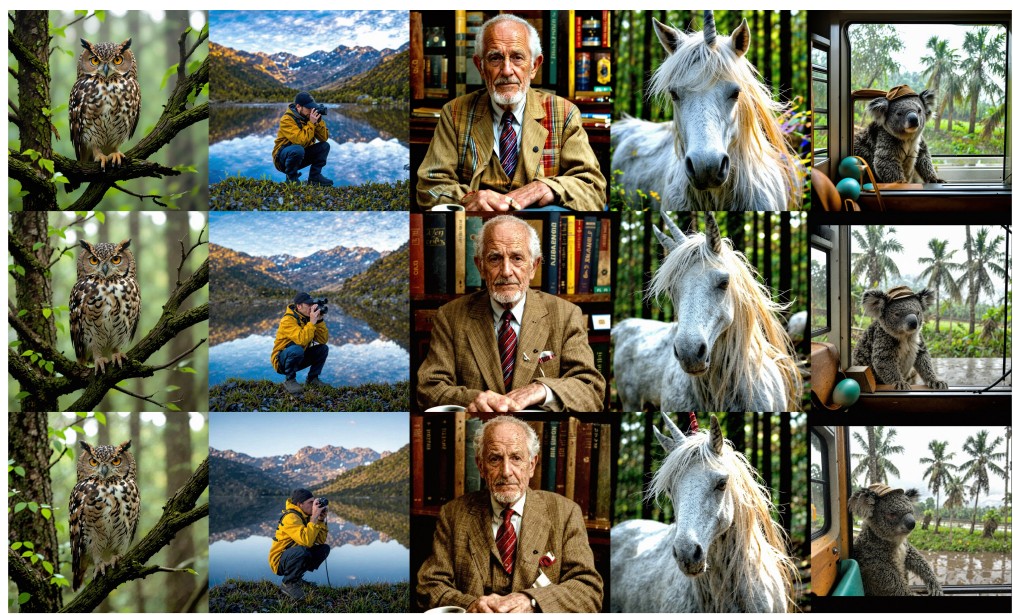

Figure 8: Effect of IDA. Adding IDA improves global fidelity and reduces artifacts across all step counts while preserving step consistency: sharper textures, cleaner edges, and more coherent object geometry are observed at 4/8/16 steps.

**Setup.** We distill an SD 3.5 Medium (2B) teacher into a 4-step student $G_\theta$. Unless otherwise noted, we use Euler integration and keep prompts and latent seeds fixed across step counts. For each column we show three rows: *top* = 4 steps, *middle* = 8 steps, *bottom* = 16 steps. Fig. 7 reports *DMD + VFM discriminator* at TTUR 5:1; Fig. 8 reports *DMD + IDA + VFM discriminator* at the same TTUR.

**Observation A (Step consistency ⇒ bounded self-consistent error).** Across all columns in Fig. 7 and Fig. 8, the 4/8/16-step samples remain close. This indicates that the distilled student follows a smooth velocity field along the FM schedule. In our notation (Eq. 20), this is *consistent with* (and provides empirical support for) the generator's *self-consistent error* $\bar{\beta}_k$ being *bounded* throughout training (i.e., the learned field does not exhibit unstable behavior between anchor timesteps). As a sanity check, if the learned field were highly inconsistent between anchors (e.g., large/drifting mismatch to its pathwise FM target), then refining the Euler discretization (4→16) would typically induce noticeable trajectory drift and different endpoints, which is not observed here. Consequently, the non-divergent $\varepsilon$ in Proposition A.7 is empirically supported.

**Observation B (IDA improves image quality while retaining consistency).** Comparing Fig. 7 (no IDA) with Fig. 8 (with IDA), IDA consistently enhances image quality at all step counts—fewer artifacts, more faithful textures, and more coherent structures—while maintaining the same level of step consistency. This matches the theory: IDA reduces inner-loop drift by keeping the fake model close to the generator (thereby stabilizing tracking, i.e., reducing $\tilde{d}_k$), and together with bounded self-consistency (i.e., controlled $\tilde{\beta}_k$ as supported above), yields a stable $\varepsilon$-best-response guarantee. Substituting $\mathbb{E}_t D_{\mathrm{KL}}(p_g\|p_f) \leq \varepsilon$ into Proposition A.2 yields the sandwich bound in Eq. 30, implying that the outer loop approximately minimizes $\mathbb{E}_t \mathrm{KL}(p_g\|p_r)$ within $\varepsilon$.

# B  IMPLEMENTATION DETAILS AND MORE EXPERIMENTAL RESULTS

## B.1  IMPLEMENTATION DETAILS

Our entire framework is implemented in PyTorch with CUDA acceleration and is trained using 8 A100 GPUs with a total batch size of 8. We adopt the AdamW optimizer (Loshchilov, 2017) with hyperparameters $\beta_1 = 0.9$ and $\beta_2 = 0.999$. The learning rate is set to $1e-6$ for the distillation of SDXL and SD 3.5 Large, and $1e-5$ for FLUX.1 dev. To efficiently support large-scale model training, we utilize Fully Sharded Data Parallel (FSDP), which enables memory-efficient and scalable training of large models.

**Timestep settings.** We adopt different coarse timestep schedules depending on the model architecture. For SDXL, we follow the 1000-step discrete DDPM schedule used in DMD2 (Yin et al., 2024a), selecting step indices $\{249, 499, 749, 999\}$. For SD 3.5 Large, we switch to continuous timestep values $\{0.25, 0.5, 0.75, 1.0\}$, which are more suitable for flow-based models. In the case of FLUX.1 dev, which adopts a shifted $\sigma$ inference strategy, we directly use the corresponding sigmas $\{0.512, 0.759, 0.904, 1.0\}$ as coarse anchors.

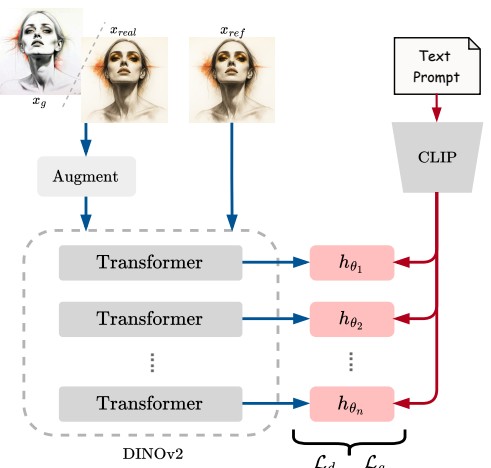

Figure 9: Design of the VFM-based discriminator.

**Training details.** We set the default TTUR (Two Time-Scale Update Rule) ratio to 5 in our main experiments on SDXL, SD 3.5 Large, and FLUX.1 dev. For large flow-based models such as SD 3.5 Large and FLUX.1 dev, we apply all proposed improvements, including Implicit Distribution Alignment (IDA), Intra-Segment Guidance (ISG), and the VFM-based Discriminator. For the diffusion-based SDXL model, we employ ISG and the VFM-based Discriminator while omitting IDA.

## B.2  DETAILED VFM-BASED DISCRIMINATOR DESIGN

As shown in Fig. 9, the discriminator integrates pretrained vision (DINOv2) and language (CLIP) encoders to provide semantically rich and spatially aligned supervision. Given an input image $x$, we

apply normalization (from $[-1, 1]$ to $[0, 1]$) and differentiable data augmentation (including color jitter, translation, and cutout). The augmented image is processed by a frozen DINOv2 vision transformer to extract multi-level semantic features. Each selected layer output is reshaped into a 2D spatial map (e.g., $[B, C, H, W]$) and passed through a lightweight convolutional head composed of spectral-normalized residual blocks.

A reference image $x_{\text{ref}}$ is processed through the same DINOv2 pathway (without augmentation) to extract corresponding semantic features. Meanwhile, the text prompt is encoded by a CLIP (ViT-L/14) text encoder into a condition feature $c$, which is projected to a spatial map. Each discriminator head fuses the image feature, reference feature, and prompt condition via element-wise multiplication and spatial summation to compute the final logits. (Note: In Section 3.4, we described the reference features $r$ as extracted by the CLIP encoder. In practice, $r = f_{\text{VFM}}(x_{\text{ref}})$ is obtained using the same DINOv2 backbone as the input image. The Fig. 2 should also be corrected.)

## B.3 TRAINING-TIME OVERHEAD OF ISG AND IDA

Under TTUR with frequency $f{=}5$, both components are applied only on generator updates (every $f$ iterations), which amortizes their cost. We measure wall-clock time (minutes) at multiple checkpoints and report the average per-iteration time (seconds/iter). All runs use $8{\times}$A100. Total training steps and times: SDXL 18k (32 hours), SD 3.5 Large 27k (56.3 hours), FLUX.1 dev 12k (23.4 hours). We report relative overhead as

$$\%\text{Overhead} = \frac{t_{\text{with}} - t_{\text{base}}}{t_{\text{base}}} \times 100\%.$$

Table 3: Training-time overhead of ISG and IDA.

| Method / Time (min) | 3k | 6k | 9k | 12k | 15k | Avg. time (s)/iter | % Overhead |
|---|---|---|---|---|---|---|---|
| Ours SDXL | 313 | 638 | 968 | 1284 | 1600 | 6.400 | **+4.44%** |
| Ours SDXL w/o ISG | 310 | 621 | 925 | 1228 | 1835 | 6.128 | – |
| Ours FLUX | 352 | 705 | 1056 | 1407 | 1757 | 7.028 | **+3.23%** |
| Ours FLUX w/o ISG | 337 | 680 | 1022 | 1364 | 1702 | 6.808 | **+3.97%** |
| Ours FLUX w/o ISG, w/o IDA | 323 | 648 | 975 | 1306 | 1637 | 6.548 | – |
| Ours SD 3.5 | 375 | 750 | 1126 | 1503 | 1878 | 7.512 | **+6.16%** |
| Ours SD 3.5 w/o ISG | 355 | 720 | 1064 | 1417 | 1769 | 7.076 | **+0.57%** |
| Ours SD 3.5 w/o ISG, w/o IDA | 354 | 706 | 1059 | 1410 | 1759 | 7.036 | – |

Across backbones, enabling **ISG** increases per-iteration time by +4.44% on SDXL, +3.23% on FLUX.1 dev, and +6.16% on SD 3.5 Large. Adding **IDA** on top of an ISG-free baseline adds +3.97% on FLUX and +0.57% on SD 3.5. These overheads of ISG (3–6%) and IDA (0.6–4%) are minor and manageable in practice, especially because both are executed only on generator steps under TTUR. In return, they bring consistent gains in convergence stability and sample quality.

## B.4 DETAILED RESULTS AND DISCUSSION OF GENEVAL AND T2I-COMPBENCH

Tables 4 and 5 report detailed compositional results. On *SD 3.5*, our 4-step models (same distilled weights, different samplers) achieve the strongest overall performance: the Euler variant attains the highest GenEval among 4-step methods (Tab. 4) and is best on five of the six T2I-CompBench categories while ranking second on Spatial (Tab. 5). This shows that our distillation preserves fidelity while improving fine-grained attribute binding and multi-object reasoning.

On *SDXL*, our distilled model leads GenEval among SDXL distillation methods and is best on Color/Shape/Texture in T2I-CompBench, while remaining competitive on Spatial, Non-spatial and Complex-3-in-1. These gains, together with strong human-preference metrics in the main text, indicate better semantic faithfulness at the same 4-step budget.

On *FLUX*, using the same distilled weights, the two solvers present a complementary trade-off: the stochastic sampler attains the highest GenEval among 4-step baselines, whereas the Euler variant is best on Shape and second on Texture/Spatial/Complex-3-in-1 in T2I-CompBench. Overall, the

two compositional suites corroborate our COCO-5K findings: the proposed distillation improves semantic alignment and compositional correctness across diverse teachers without increasing the sampling cost.

Table 4: GenEval benchmark results. We report overall score and per-attribute pass rates (%). **Bold**/Underline: best/second best in distilling the same teacher.

| Method | #NFE | GenEval↑ | Single Obj. | Two Obj. | Counting | Colors | Position | Color-Attr. |
|---|---|---|---|---|---|---|---|---|
| **SDXL Distillation** | | | | | | | | |
| LCM–SDXL | 8 | 0.5036 | 99.06% | 55.56% | 39.69% | 85.37% | 7.00% | 15.50% |
| PCM–SDXL | 4 | 0.4944 | 97.50% | 56.06% | 39.69% | 81.65% | 7.50% | 14.25% |
| SDXL–Lightning | 4 | 0.5332 | 98.44% | 60.35% | 45.31% | 84.57% | 10.50% | 20.75% |
| Hyper–SDXL | 4 | 0.5398 | 98.44% | 65.40% | 38.75% | 88.83% | 12.50% | 20.00% |
| DMD2–SDXL | 4 | 0.5779 | 99.69% | 75.76% | 47.81% | 87.50% | 10.50% | 25.50% |
| Ours–SDXL | 4 | **0.5784** | 99.69% | 73.74% | 47.81% | 88.83% | 10.00% | 27.00% |
| **SD 3.5 Large Distillation** | | | | | | | | |
| SD 3.5 Large Turbo | 4 | 0.6877 | 99.06% | 88.89% | 68.75% | 77.93% | 23.00% | 55.00% |
| Ours–SD 3.5 | 4 | 0.6955 | 99.06% | 92.93% | 63.44% | 81.12% | 22.00% | 58.75% |
| Ours–SD 3.5 (Euler) | 4 | **0.7098** | 100.00% | 91.67% | 67.81% | 81.38% | 24.50% | 60.50% |
| **FLUX Distillation** | | | | | | | | |
| FLUX.1 schnell | 4 | 0.6807 | 99.38% | 89.39% | 60.00% | 77.93% | 29.00% | 52.75% |
| Hyper–FLUX | 4 | 0.6193 | 98.12% | 69.95% | 67.50% | 75.53% | 16.75% | 43.75% |
| FLUX–Turbo–Alpha | 4 | 0.4724 | 88.12% | 44.70% | 52.50% | 64.63% | 13.25% | 20.25% |
| Ours–FLUX | 4 | **0.6471** | 98.75% | 70.71% | 82.50% | 80.05% | 14.00% | 42.25% |
| Ours–FLUX (Euler) | 4 | 0.6420 | 99.06% | 71.91% | 80.00% | 78.72% | 16.50% | 40.50% |
| **Teachers** | | | | | | | | |
| SDXL | 80 | 0.5461 | 96.88% | 69.70% | 41.88% | 87.23% | 10.25% | 21.75% |
| SD 3.5 Large | 80 | 0.7140 | 100.00% | 90.66% | 69.38% | 81.38% | 26.50% | 60.50% |
| FLUX.1 Dev | 50 | 0.6689 | 99.38% | 82.83% | 74.06% | 77.66% | 22.00% | 46.00% |
| FLUX.1 Dev | 25 | 0.6733 | 99.69% | 84.34% | 75.31% | 81.12% | 20.75% | 42.75% |

## B.5 ANALYSIS OF DIVERSITY

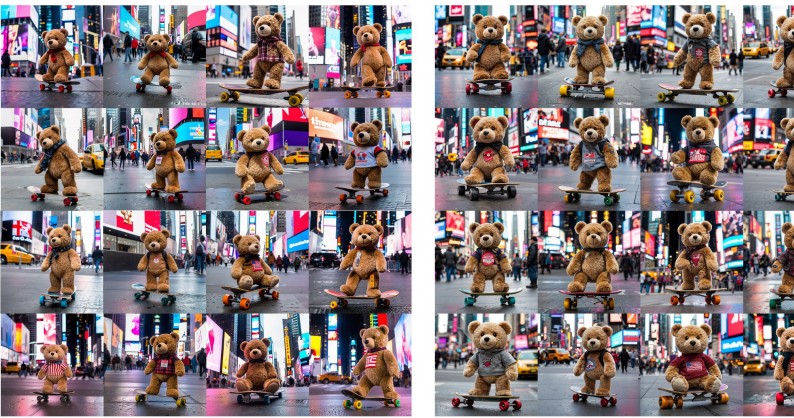

Figure 10: Qualitative diversity under the VFM discriminator. Left: DMD2-SDXL. Right: Ours-SDXL.

**Setup.** To quantify the impact of the VFM discriminator on sample diversity, we construct a prompt validation set and, for each prompt, generate 16 samples per model. On this set, we compute two diversity metrics across all image pairs: (i) *LPIPS-diversity*, defined as the average pairwise LPIPS distance between samples; and (ii) *CLIP-diversity*, obtained by feeding all images through a pre-

Table 5: 4-Step Results on T2I-CompBench. **Bold**/Underline: best/second best in distilling the same teacher. Our distilled SD 3.5 model approaches overall state-of-the-art distillation performance.

| Method | Color | Shape | Texture | Spatial | Non-spatial | Complex-3-in-1 |
|---|---|---|---|---|---|---|
| LCM-SDXL | 0.5997 | 0.4015 | 0.4958 | 0.1672 | 0.3010 | 0.3364 |
| SDXL-Lightning | 0.5758 | 0.4492 | 0.5154 | 0.2124 | 0.3098 | 0.3517 |
| Hyper-SDXL | 0.6435 | 0.4732 | 0.5581 | 0.2213 | **0.3104** | 0.3301 |
| PCM-SDXL | 0.5591 | 0.4142 | 0.4693 | 0.2013 | 0.3099 | 0.3234 |
| DMD2-SDXL | 0.6531 | 0.4816 | 0.5967 | **0.2231** | 0.3100 | **0.3597** |
| Ours-SDXL | **0.6867** | **0.4828** | **0.5989** | 0.2224 | 0.3100 | 0.3594 |
| SD 3.5 Large Turbo | 0.7050 | 0.5443 | 0.6512 | 0.2839 | 0.3130 | 0.3520 |
| Ours-SD 3.5 | 0.7657 | 0.6069 | 0.7427 | **0.2970** | 0.3177 | 0.3916 |
| Ours-SD 3.5 (Euler) | **0.7711** | **0.6149** | **0.7543** | 0.2857 | **0.3182** | **0.3968** |
| FLUX.1 schnell | 0.7317 | 0.5649 | 0.6919 | 0.2626 | 0.3122 | 0.3669 |
| Hyper-FLUX | **0.7465** | 0.5023 | **0.6153** | **0.2945** | **0.3116** | **0.3766** |
| FLUX-Turbo-Alpha | 0.7406 | 0.4873 | 0.6024 | 0.2501 | 0.3094 | 0.3688 |
| Ours-FLUX | 0.7284 | 0.5055 | 0.6031 | 0.2451 | 0.3028 | 0.3652 |
| Ours-FLUX (Euler) | 0.7363 | **0.5120** | 0.6112 | 0.2521 | 0.3028 | 0.3697 |

Table 6: LPIPS-diversity (average pairwise LPIPS distance) and CLIP-diversity (variance in CLIP image embeddings) on a 23-prompt validation set with 16 samples per prompt. Higher is better.

| Method | LPIPS-Diversity ↑ | CLIP-Diversity ↑ |
|---|---|---|
| DMD2-SDXL | 0.5960 | 0.0985 |
| Ours-SDXL | 0.6002 | 0.0802 |
| SD 3.5 Large Turbo | 0.5659 | 0.0879 |
| Ours SD 3.5 Large | 0.5664 | 0.0900 |

trained CLIP image encoder, $\ell_2$-normalizing the features, and averaging $1 - \text{cosine-similarity}$ over all pairs. Both metrics are averaged over prompts, and higher values indicate higher sample diversity.

**Results.** The quantitative results are shown in Tab. 6. For SDXL, Ours-SDXL achieves LPIPS-diversity **0.6002** vs. **0.5960** for the DMD2-SDXL baseline, and CLIP-diversity **0.0802** vs. **0.0985**. For SD 3.5 Large, our method obtains LPIPS-diversity **0.5664** vs. **0.5659** for the SD 3.5-Turbo baseline and CLIP-diversity **0.0900** vs. **0.0879**. Overall, LPIPS-based diversity remains essentially unchanged (all methods within about 1-2%), while CLIP-diversity changes mildly: it decreases slightly for SDXL ($0.0985 \rightarrow 0.0802$). This supports our original claim that the VFM discriminator slightly reshapes the distilled generator toward human-preferred semantic modes; and diversity change is small compared to the gains in human-preference-aligned metrics.

**Qualitative visualization.** To further illustrate this effect, in Fig. 10, we visualize 16 samples for the prompt "*a teddy bear on a skateboard in Times Square*" for DMD2-SDXL (left) and Ours-SDXL (right). Both models exhibit clear variation in pose, background, and apparel. DMD2-SDXL shows slightly more variation in viewpoint (e.g., more side-angle shots of the bear), while our model tends to produce more centered, compositionally consistent bears—matching the mild CLIP-diversity difference and the improved human-preference performance reported in Tab. 2.

In summary, the VFM discriminator gently reshapes the distilled generator toward human-preferred, semantically meaningful modes, while inducing only a mild change in standard diversity measures.

## B.6 1–2-STEP GENERATION

To assess the behavior of SenseFlow beyond the 4-step regime, we further evaluate 2-step and 1-step generation across SDXL, SD 3.5 Large, and FLUX. The results are summarized in Tab. 7.

For 2-step sampling, SenseFlow naturally extends without any architectural modification and consistently outperforms strong baselines under the same setup and evaluation protocol. On SDXL, the 2-step model surpasses DMD2-SDXL on all human-preference metrics (HPSv2, AESv2, PickScore, ImageReward). On SD 3.5 Large, the improvement is particularly pronounced: at 2 steps, the model

Table 7: SenseFlow in the low-NFE regime (2-step and 1-step). We compare the distilled students with corresponding baselines on SDXL, SD 3.5, and FLUX.

| Method | #NFE | CLIP ↑ | HPSv2 ↑ | AESv2 ↑ | PickScore ↑ | ImageReward ↑ |
|---|---|---|---|---|---|---|
| DMD2-SDXL | 2 | 0.3295 | 0.2813 | 5.432 | 22.57 | 0.8666 |
| Ours-SDXL | 2 | 0.3263 | **0.2827** | **5.710** | **22.89** | **0.9192** |
| Ours-SDXL | 1 | **0.3298** | 0.2818 | 5.584 | 22.24 | 0.8570 |
| SD3.5-Large-Turbo | 2 | 0.3248 | 0.2745 | 5.394 | 22.16 | 0.7188 |
| Ours-SD3.5 Large | 2 | 0.3326 | **0.2889** | **5.537** | **22.88** | **1.2022** |
| Ours-SD3.5 Large | 1 | **0.3332** | 0.2803 | 5.421 | 22.32 | 1.0651 |
| Hyper-FLUX | 2 | 0.3176 | 0.2682 | 5.541 | 21.63 | 0.3636 |
| Ours-FLUX | 2 | **0.3207** | **0.2866** | **5.926** | **22.38** | **0.9296** |

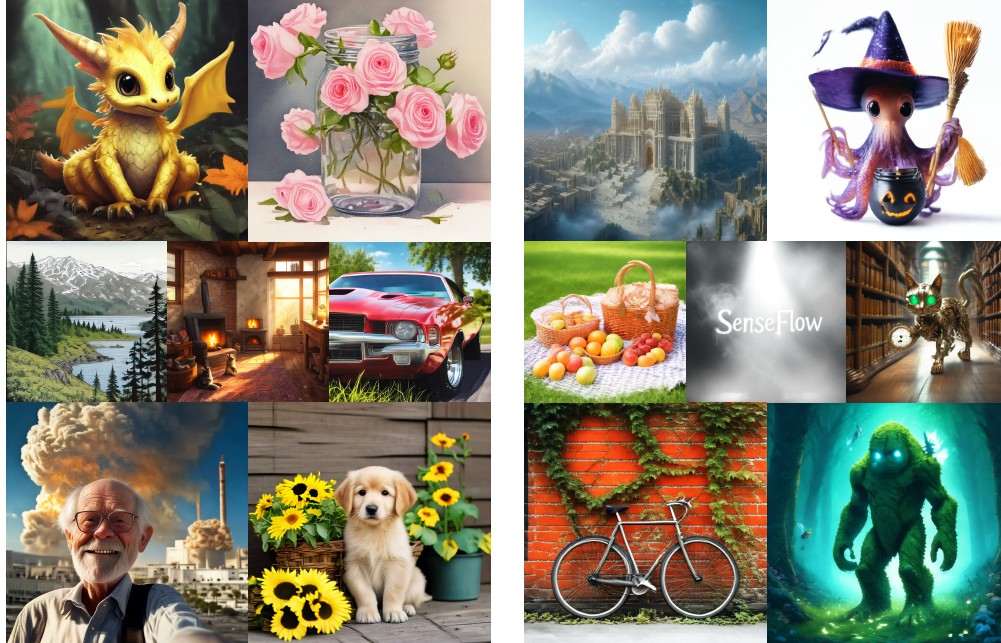

Figure 11: 1-step 1024×1024 samples produced by our SenseFlow on SDXL (Left) and SD 3.5 Large (Right).

achieves an ImageReward of **1.2022** and a PickScore of **22.88**, clearly exceeding SD 3.5-Large-Turbo. For FLUX, the 2-step SenseFlow student also improves over Hyper-FLUX on all metrics.

For 1-step generation, we start from the pretrained 4-step SenseFlow students and apply a short fine-tuning schedule (6000 iterations) for the single-step setting. This yields competitive performance with only a modest drop relative to the 2-step case; for example, the 1-step SD 3.5 Large model attains an ImageReward of **1.0651** and a CLIP score of **0.3332**. Qualitative comparisons for 2-step and 1-step sampling are provided in the Fig. 15 and Fig. 11. In particular, for SD 3.5 Large, the visual degradation from 4 steps to 2 steps is very small (from columns 3 to 5 of Fig. 15), indicating that SenseFlow remains robust even in aggressive low-NFE regimes.

Overall, these results demonstrate that SenseFlow delivers strong generation quality at 2 steps without architectural changes and exhibits promising single-step performance after light tuning. This highlights its potential to further advance 1–2-step distillation for large (8B+) text-to-image models, a setting where existing open-source solutions remain limited.

Table 8: Effect of ISG on training dynamics (SD 3.5 Large). FID-T on COCO-5k at different training iterations for the full method (with ISG) and an ablation without ISG. Lower is better.

| Method | 1.5k | 3k | 4.5k | 6k | 9k | 12k |
|---|---|---|---|---|---|---|
| Ours (w/ ISG) | 14.48 | 22.32 | 17.65 | 18.11 | 16.18 | 15.20 |
| Ours w/o ISG | 138.2 | 24.99 | 19.43 | 19.12 | 18.07 | 19.06 |

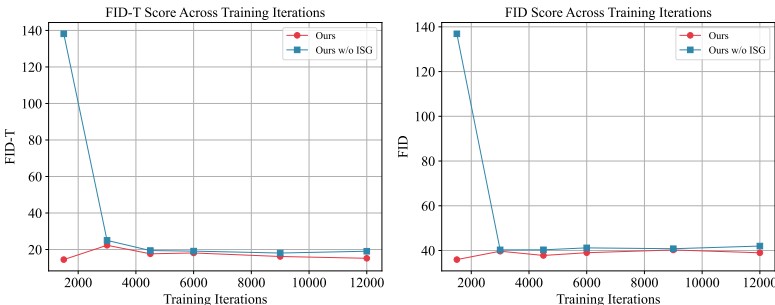

Figure 12: FID-T and FID scores across training iterations on COCO-5k for SD 3.5 Large, comparing the full method with ISG and an ablation without ISG.

## B.7 IMPACT OF ISG ON TRAINING DYNAMICS

We study the effect of Intra-Segment Guidance (ISG) on training stability and convergence speed for SD 3.5 Large. Fig. 12 visualizes the evolution of FID and FID-T on COCO-5k as a function of training iterations for the full method (with ISG) and an ablation without ISG, while the corresponding FID-T values at selected checkpoints are summarized in Tab. 8.

**Stability and convergence speed.** ISG mainly improves the *stability* and *convergence speed* of training. At 1.5k iterations, the model trained without ISG still exhibits extremely poor alignment to the teacher (FID-T $\approx$ **138**), whereas the model with ISG has already reached a reasonable regime (FID-T $\approx$ **14.48**). As training proceeds, both variants become more stable, but the ISG model consistently attains lower FID-T throughout training (e.g., 16.18 vs. 18.07 at 9k iterations, and 15.20 vs. 19.06 at 12k iterations). A similar gap is observed on the FID curves in Fig. 12. These results indicate that ISG effectively redistributes timestep importance within each segment, making the student updates less sensitive to under-trained timesteps and leading to faster and more stable convergence.

## B.8 COMPARISON TO DMD2 AND DETAILED COMPONENT ABLATIONS

**On DMD2 for SD 3.5 Large.** For SDXL, we directly use the official DMD2-SDXL model as a strong baseline in the main table. For SD 3.5 Large, however, there is currently no public DMD2 implementation. We therefore implement a faithful DMD2-style latent discriminator on top of the DiT backbone and attempt to train a "DMD2-SD 3.5 Large" student under the same settings. Despite extensive tuning (e.g., increasing the TTUR up to 20), this model fails to converge in practice and collapses to almost all-black samples. Only after replacing the vanilla discriminator with the proposed VFM-based discriminator does training start to produce meaningful images, albeit still with noticeable instability. Adding IDA and ISG on top of VFM then gradually improves the quality and stability, eventually yielding the performance reported for SenseFlow.

Because the DMD2-SD 3.5 Large variant does not reach a meaningful operating regime (i.e., it collapses), we do not include it as a baseline in Tab. 1. Instead, the we provide qualitative visualizations of this progression in Fig. 13: from DMD2 (black images) to DMD2+VFM, then DMD2+VFM+IDA (Ours w/o ISG), Ours (full), and DMD2+VFM+ISG (Ours w/o IDA). This visual sequence illustrates why a direct DMD2 baseline is difficult to scale to SD 3.5 Large and how the proposed modifications mitigate its failure modes.

**More comprehensive ablations.** To analyze the contribution of individual components, we include an expanded ablation in Tab. 2. For SD 3.5 Large, the table reports: *(a)* the full SenseFlow model, *(b)* a variant without ISG (w/o ISG), *(c)* a variant without IDA (w/o IDA), and *(d)* a variant without both ISG and IDA (w/o ISG, w/o IDA). Performance degrades steadily along this path: removing ISG already causes a noticeable drop across FID-T and human-preference metrics, removing IDA leads to further degradation, and removing both ISG and IDA results in a clear collapse (FID-T increases from 13.38 to 43.84, with large drops in HPSv2, PickScore, and ImageReward). This pattern confirms that both IDA and ISG contribute substantially to the final performance and that they interact synergistically.

For SDXL, the same table compares DMD2-SDXL, DMD2 equipped with the VFM discriminator (DMD2 w VFM), and the full Ours-SDXL model. This isolates the effect of the VFM discriminator (DMD2 vs. DMD2 w VFM) and then the additional gains obtained by integrating ISG on top (DMD2 w VFM vs. Ours-SDXL). In particular, equipping DMD2 with VFM improves most human-preference metrics, and adding the full SenseFlow pipeline yields further consistent gains.

Overall, Tab. 9 and the accompanying qualitative grid in Fig. 13 together provide a step-by-step view of how the three components—VFM discriminator, IDA, and ISG—each improve the model and how their combination yields the strongest overall performance, while also clarifying why a direct DMD2 baseline is not informative for SD 3.5 Large.

Table 9: Expanded ablation study of IDA, ISG, and the VFM discriminator.

| Method | FID-T↓ | HPSv2↑ | Pick↑ | ImageReward↑ | AESv2↑ |
|---|---|---|---|---|---|
| **Stable Diffusion 3.5 Large** | | | | | |
| Ours | **13.38** | **0.3015** | **23.03** | **1.1713** | **5.482** |
| w/o ISG | 17.00 | 0.2971 | 22.75 | 1.0186 | 5.453 |
| w/o IDA | 17.83 | 0.2800 | 22.47 | 0.9365 | 5.407 |
| w/o ISG, w/o IDA | 43.84 | 0.2555 | 20.60 | 0.3828 | 5.102 |
| **Stable Diffusion XL** | | | | | |
| DMD2-SDXL | **15.04** | 0.2964 | 22.98 | 0.9324 | 5.530 |
| DMD2 w VFM | 18.55 | 0.2995 | 23.00 | 0.9744 | 5.625 |
| Ours-SDXL | 17.76 | **0.3010** | **23.17** | **0.9951** | **5.703** |

Table 10: Quantitative results of different adversarial loss weights.

| Method | FID-T ↓ | CLIP Score ↑ | HPSv2 ↑ | PickScore ↑ | ImageReward ↑ |
|---|---|---|---|---|---|
| Hyper-SDXL | **13.71** | **0.3254** | 0.3000 | 22.98 | 0.9777 |
| Ours ($\lambda_G = 0.25$) | 17.53 | 0.3234 | 0.3003 | 23.15 | 0.9326 |
| Ours ($\lambda_G = 0.5$) | 17.76 | 0.3248 | **0.3010** | **23.17** | **0.9951** |

Table 11: Quantitative results of different backbone scales.

| Method | FID-T ↓ | CLIP Score ↑ | HPSv2 ↑ | PickScore ↑ | ImageReward ↑ |
|---|---|---|---|---|---|
| Ours w ViT-S | 17.26 | **0.3262** | 0.2983 | 23.12 | **0.9635** |
| Ours w ViT-B | **16.58** | 0.3234 | 0.2991 | 23.07 | 0.9218 |
| Ours w ViT-L | 17.53 | 0.3239 | **0.3003** | **23.15** | 0.9326 |

## B.9 Qualitative comparisons

We further provide qualitative side-by-side comparisons for SDXL, SD 3.5 Large, and FLUX. For each model family, we show samples from SenseFlow and strong baselines under the same prompts. As illustrated in Figures 14–16, SenseFlow tends to produce images with sharper details, cleaner structure, and more faithful overall quality.

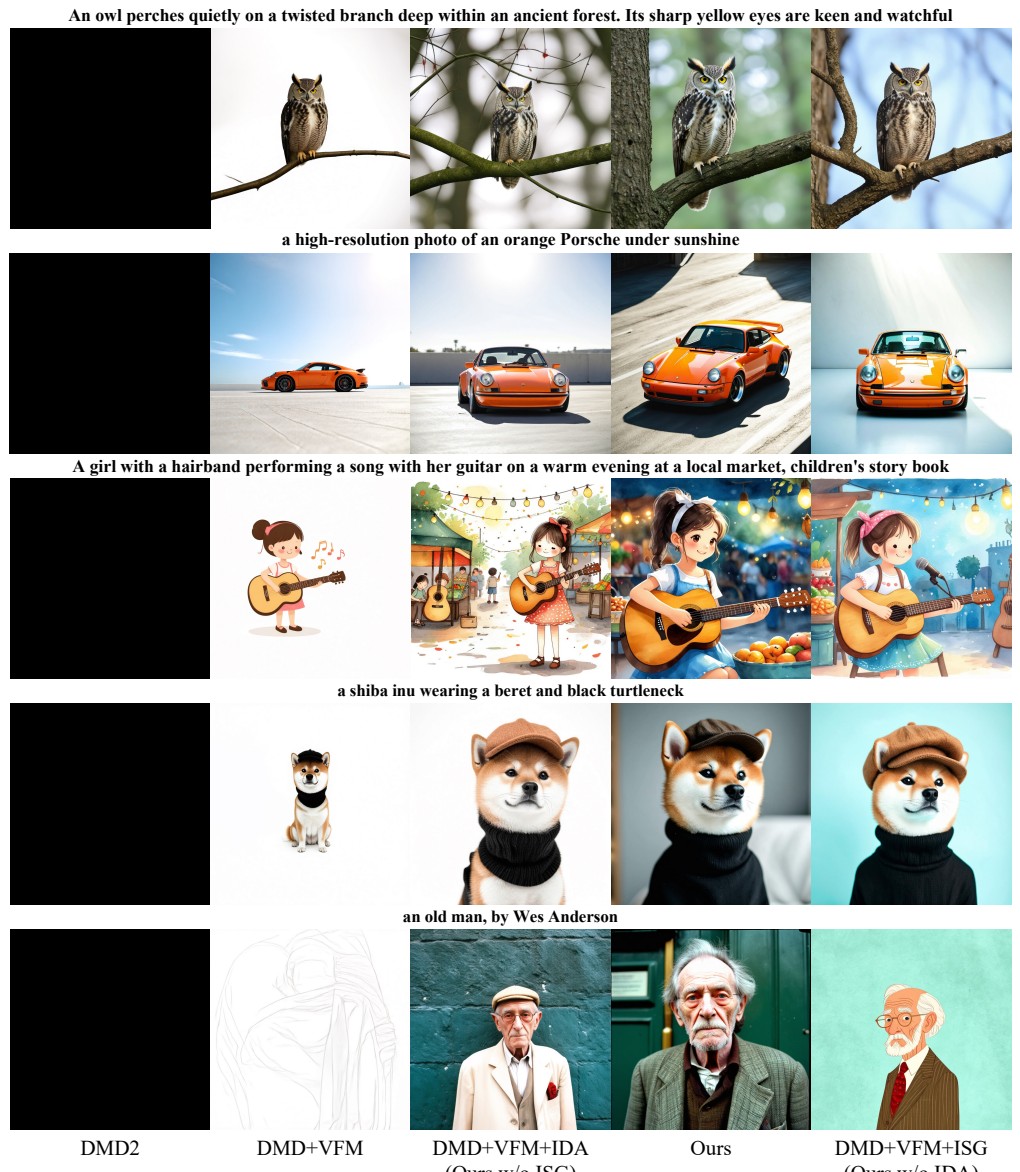

Figure 13: Qualitative ablation on SD 3.5 Large under shared prompts. Plain DMD2 collapses to all-black samples; adding VFM recovers basic structure, while introducing IDA and ISG progressively improves fidelity and text–image alignment, with the full SenseFlow model yielding the most visually pleasing and semantically faithful results..

## B.10    MORE ABLATION STUDY RESULTS AND VISUALIZATION SAMPLES

**Effect of Different Adversarial Loss Weights.** In our main experiments, the hyperparameter $\lambda_G$ in Algorithm 1, Line 22, is set to 0.5, 0.1, and 2.0 for SDXL, SD 3.5 Large, and FLUX.1 dev, respectively. To further investigate the impact of this hyperparameter, we conduct an ablation study using SDXL as an example, decreasing $\lambda_G$ to 0.25. The results are presented in Tab. 10. We observe that setting $\lambda_G = 0.5$ leads to improved performance across most metrics, including CLIP Score, HPSv2, PickScore, and ImageReward. Notably, this configuration achieves the best scores on HPSv2, PickScore, and ImageReward among all methods in Tab. 1. These results highlight the strong semantic and visual supervision capabilities of our VFM-based discriminator.

**Results of Different Backbone Scales.** We evaluate the impact of different VFM backbone scales (ViT-S, B, and L) in the discriminator on SDXL distillation. Interestingly, the results (Tab.1) do not follow a monotonic trend with respect to model size. ViT-B achieves the best FID-T, while ViT-S yields higher CLIP Score and ImageReward. ViT-L slightly outperforms others on HPSv2 and PickScore. These findings suggest that different backbone scales offer different trade-offs in semantic alignment versus visual fidelity, and that larger backbones do not necessarily guarantee consistent improvements across all metrics. This observation is partially consistent with findings in the ADD(Sauer et al., 2024b) paper, which also noted diminishing returns when scaling the discriminator. In our main paper, we adopt ViT-L as the default backbone for the VFM-based discriminator.

**More visualization results of our methods on SD 3.5 Large and SDXL.** As shown in Fig. 17 and Fig. 18, we present more samples produced by out 4-step generator distilled from SD 3.5 Large and SDXL, separately. The prompts of these samples are listed later in our appendix.

### B.11 PROMPTS FOR FIG. 1, FIG. 17, AND FIG. 18

We use the following prompts for Fig. 1. From left to right, top to bottom:

- A red fox standing alert in a snow-covered pine forest
- A girl with a hairband performing a song with her guitar on a warm evening at a local market, children's story book
- Astronaut on a camel on mars
- A cat sleeping on a windowsill with white curtains fluttering in the breeze
- A stylized digital art poster with the word "SenseFlow" written in flowing smoke from a stage spotlight
- A surreal landscape inspired by The Dark Side of the Moon, with floating clocks and rainbow beams
- a hot air balloon in shape of a heart. Grand Canyon
- A young man with a leather jacket and messy hair playing a cherry-red electric guitar on a rooftop at sunset
- A young woman wearing a denim jacket and headphones, walking past a graffiti wall
- A photographer holding a camera, squatting by a lake, capturing the reflection of the mountains in an early morning
- a young girl playing piano
- A close-up of a woman's face, lit by the soft glow of a neon sign in a dimly lit, retro diner, hinting at a narrative of longing and nostalgia

Besides, we use the following prompts for Fig. 17. From left to right, top to bottom:

- A quiet room with Oasis album covers framed on the wall, acoustic guitar resting on a stool
- An astronaut lying in the middle of white ROSES, in the style of Unsplash photography.
- cartoon dog sits at a table, coffee mug on hand, as a room goes up in flames. "Help" the dog is yelling
- Art illustration, sports minimalism style, fuzzy form, black cat and white cat, solid color background, close-up, pure flat illustration, extreme high-definition picture, cat's eyes depict clear and meticulous, high aesthetic feeling, graphic, fuzzy, felt, minimalism, blank space, artistic conception, advanced, masterpiece, minimalism, fuzzy fur texture.
- Close-up of the top peak of Aconcagua, a snow-covered mountain in the Himalayas at sunrise during the golden hour. Award-winning photography, shot on a Canon EOS R5 in the style of Ansel Adams.
- A curvy timber house near a sea, designed by Zaha Hadid, represents the image of a cold, modern architecture, at night, white lighting, highly detailed
- a teddy bear on a skateboard in times square

- a black and white picture of a woman looking through the window, in the style of Duffy Sheridan, Anna Razumovskaya, smooth and shiny, wavy, Patrick Demarchelier, album covers, lush and detailed

As for Fig. 18, we use following prompts from left to right, top to bottom:

- Astronaut in a jungle, cold color palette, muted colors, detailed, 8k
- A bookshelf filled with colorful books, a potted plant, and a small table lamp
- A dreamy beachside bar at dusk serving mojitos and old fashioneds, guitars hanging on the wall
- A portrait of a human growing colorful flowers from her hair. Hyperrealistic oil painting. Intricate details.
- Peach-faced lovebird with a slick pompadour.
- a stunning and luxurious bedroom carved into a rocky mountainside seamlessly blending nature with modern design with a plush earth-toned bed textured stone walls circular fireplace massive uniquely shaped window framing snow-capped mountains dense forests
- An acoustic jam session in a small café, handwritten setlist on the wall, cocktails on every table
- a blue Porsche 356 parked in front of a yellow brick wall.

**A towering jellyfish queen glides gracefully through an underwater kingdom, her translucent tendrils trailing behind her like an elegant gown. Bioluminescent patterns ripple across her ethereal body, pulsing in sync with the deep ocean currents. Tiny fish swim in mesmerizing formations around her, drawn to the soft, hypnotic glow that follows her every movement.**

**Digital art of a young girl reading in an enchanted forest, inspired by Arthur Rackham's whimsical style. Warm sunlight filters through lush greenery (soft greens, browns, golden yellows). A 35mm medium shot centers her, book in hand, surrounded by fairies, glowing fireflies, and curious foxes. Delicate glitter accents add magic—perfect for a child's dreamy escape.**

**Aerial shot of stunning greek fantasy palace with lots of towers and spires, large windows and airy quality and feel, palace is in the center of a sprawling city full of libraries, shops, and bustling activity, background is a rolling mountain range, large fluffy clouds in the skyUse ethereal lighting and light colors to create an idyllic atmosphere, Epic sky, Otherworldly, Hyper realistic**

**Bokeh, Electric Colors, Accent Lighting, Lightning, Inferno, insanely detailed and intricate, hypermaximalist, elegant, ornate, hyper realistic, super detailed, phoenix with wings of flame,front view,magical atmosphere.**

**A sketchbook page filled with clothing designs, with the brand name "SENSEFLOW" written in the top corner**

**In the heart of an ancient cathedral, Excalibur rests upon an altar of marble, encased in shimmering, ethereal light. The stained-glass windows cast multicolored beams across the blade, illuminating the intricate runes carved into its steel. A quiet reverence fills the chamber for legends say that only the true king may grasp its hilt without being turned to dust.**

| SDXL | LCM-SDXL | Hyper-SDXL | DMD2-SDXL | Ours-SDXL |

Figure 14: Qualitative SDXL comparisons under shared prompts. Each row corresponds to one prompt, showing SenseFlow and strong SDXL baselines side by side.

A mischievous minion transformed into a dark side warrior, inspired by Darth Vader, stands menacingly in a dimly lit chamber. Its yellow, cylindrical body is painted matte black, with glossy red accents glowing faintly. It wears a flowing black cape, a custom helmet with sharp edges and a single menacing goggle-eye glowing red. In its hand, a tiny yet powerful red lightsaber hums with energy. The minion's expression is a mix of determination and its usual playful mischief, as if ready to wreak havoc while still being adorably chaotic. The dark background is illuminated by faint red and blue lights, evoking the ominous atmosphere of a Sith lair.

A giant, four-armed baker made entirely of gingerbread hums a deep, rumbling tune as he kneads dough in a cozy, fire-lit kitchen. His icing-swirl eyebrows lift in delight as he pulls a tray of enchanted pastries from the oven—each one shaped like a tiny, dancing creature. The warm scent of cinnamon and sugar fills the air as his candy-button eyes twinkle with pride.

Japanese style tuna sushi restaurant cartoon with soft and funny contours with 3d with white background

A dramatic black-and-white portrait of a retired footballer holding his boots in one hand and a guitar in the other

A baker pulling fresh croissants from a brick oven in a rustic bakery

A stylized digital art poster with the word "SenseFlow" written in flowing smoke from a stage spotlight

|  |  |  |  |  |
|---|---|---|---|---|
| SD 3.5 L | SD 3.5 L Turbo (4) | Ours-SD 3.5 L (4) | SD 3.5 L Turbo (2) | Ours-SD 3.5 L (2) |

Figure 15: Qualitative SD 3.5 Large comparisons under shared prompts. SenseFlow produces visually appealing images with improved composition and text–image consistency compared to SD 3.5-Large-Turbo and SD 3.5-Large.

**A mischievous minion transformed into a dark side warrior, inspired by Darth Vader, stands menacingly in a dimly lit chamber. Its yellow, cylindrical body is painted matte black, with glossy red accents glowing faintly. It wears a flowing black cape, a custom helmet with sharp edges and a single menacing goggle-eye glowing red. In its hand, a tiny yet powerful red lightsaber hums with energy. The minion's expression is a mix of determination and its usual playful mischief, as if ready to wreak havoc while still being adorably chaotic. The dark background is illuminated by faint red and blue lights, evoking the ominous atmosphere of a Sith lair.**

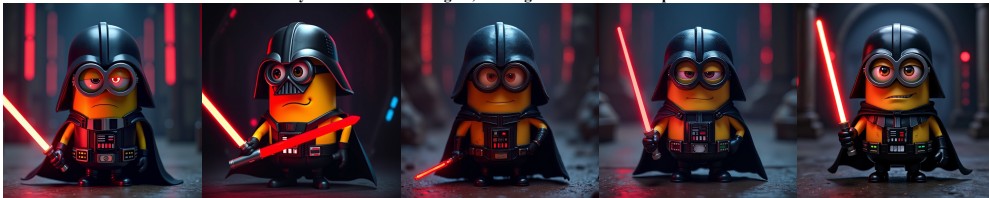

**Inside the FC Barcelona locker room, a No.10 jersey with the name "SenseFlow" hangs under the spotlight, surrounded by red and blue walls and team gear**

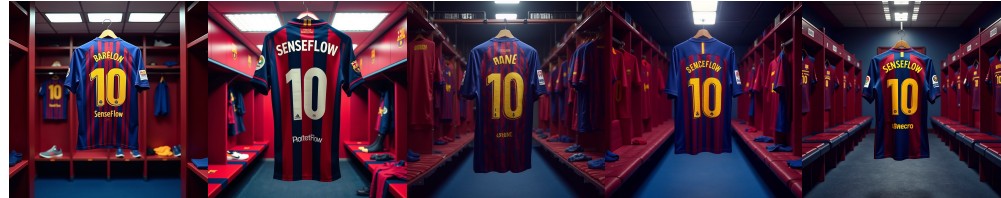

**A celestial bard with flowing, star-speckled robes strums a crystalline harp that hums with the music of the cosmos. Their silver hair drifts as if caught in an eternal breeze, and their eyes shine like twin galaxies. As they play, glowing constellations dance around them, weaving stories of forgotten legends. The air vibrates with an ethereal melody, bending reality itself to their song.**

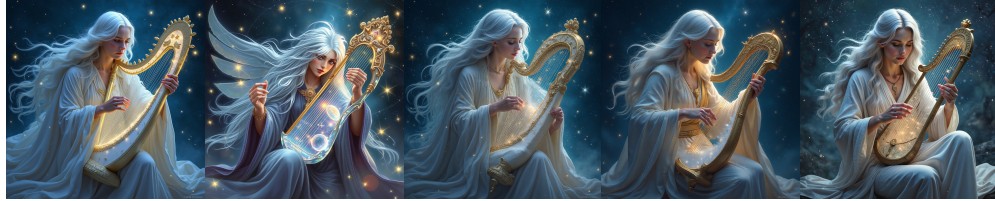

**A baker pulling fresh croissants from a brick oven in a rustic bakery**

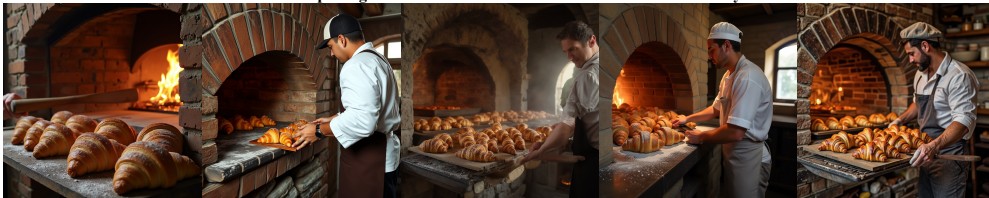

**A stylish bartender preparing a "Catalan Sunset" cocktail, with a small Barça logo etched on the glass**

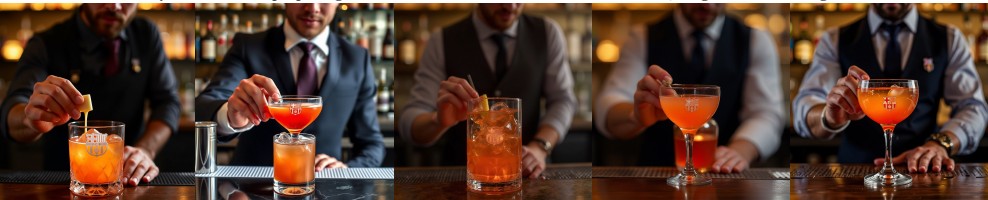

**A surreal landscape inspired by The Dark Side of the Moon, with floating clocks and rainbow beams**

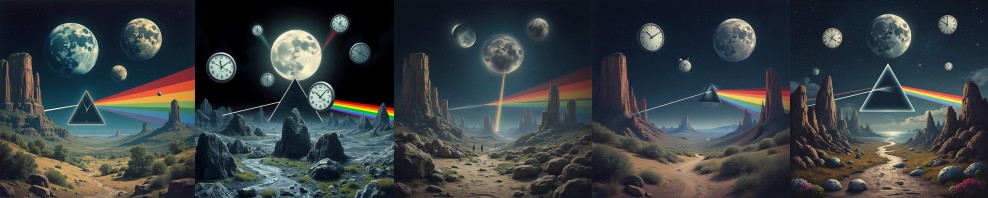

| FLUX | FLUX-Schnell | FLUX-Turbo-Alpha | Hyper-FLUX | Ours-FLUX |

Figure 16: Qualitative FLUX comparisons under shared prompts. SenseFlow provides clearer structures and better alignment with the textual descriptions than Hyper-FLUX and FLUX-Turbo-Alpha.

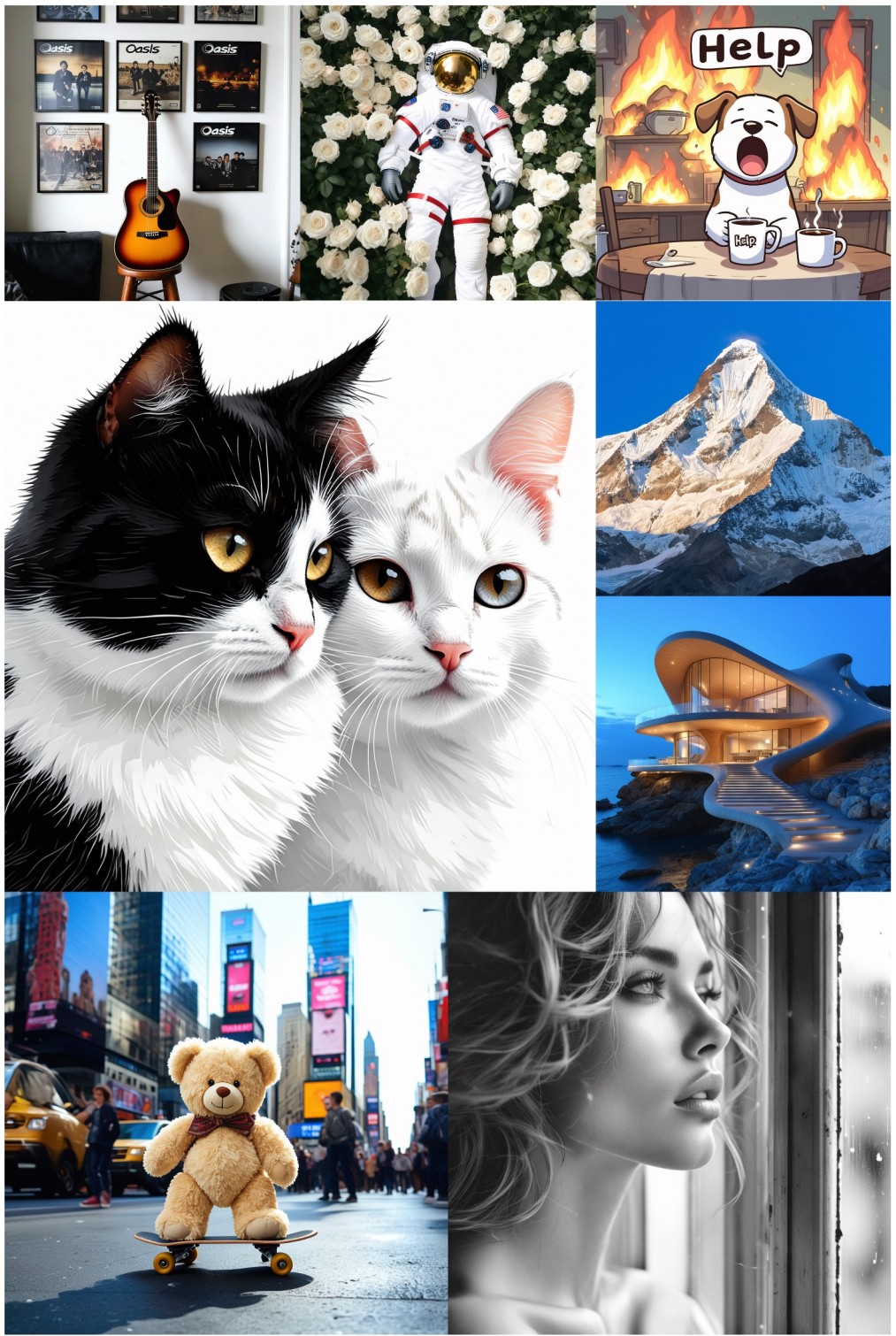

Figure 17: 1024×1024 samples produced by our 4-step generator distilled from SD 3.5 Large.

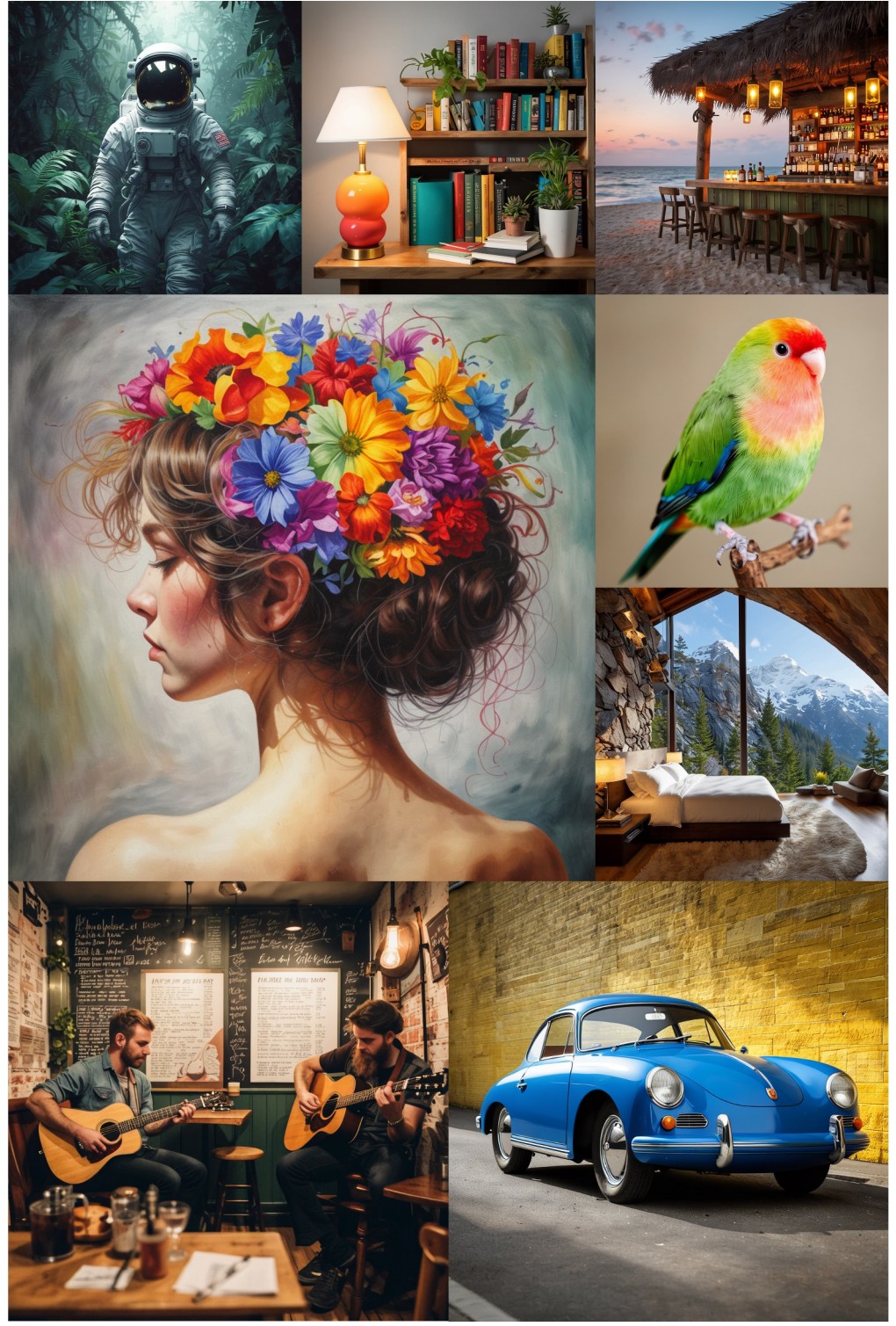

Figure 18: 1024×1024 samples produced by our 4-step generator distilled from SDXL.

