# OpenReview forum: "SenseFlow: Scaling Distribution Matching for Flow-based Text-to-Image Distillation"
_ICLR.cc/2026/Conference — ICLR 2026 Poster_

### Official Review · Reviewer_iTJr · 2025-10-19

**Soundness:** 3
**Presentation:** 3
**Contribution:** 2
**Rating:** 4
**Confidence:** 2

**Summary:**

This paper improves Distribution Matching Distillation (DMD) for large text to image models like SD 3.5 and FLUX by introducing Implicit Distribution Alignment (IDA) to handle divergence issues, Intra Segment Guidance (ISG) to improve timestep wise denoising and VFM-based discriminator for effective training.

**Strengths:**

(i) This paper makes contributions by scaling DMD to large flow-based models, such as SD3.5 and FLUX

(ii) The authors propose Implicit Distribution Alignment (IDA) is a novel method that helps stabilize training

(iii) The approach also shows strong empirical performance across multiple challenging benchmarks.

**Weaknesses:**

(i) The theoretical justification provided in Proposition 3.1 lacks clarity. Specifically, the definition of the conditional drift term $\hat{v}_{\theta}(X_t, t) = X_1 - X_0$ in line 674 is problematic. Since this quantity is conditioned on $X_1$, it should be treated as a random variable dependent on $X_1$. However, the authors appear to use it deterministically in the derivation of error bounds, which raises concerns about the correctness of the results.

(ii) The error term $\Delta_k = \mathbb{E}||v_{\phi_k} - v_{\theta_k}||$ is not well justified. According to the authors’ own note in line 667 about definition of $v_{\theta_k}$: “where the velocity field is the output of the generator network at each timestep t”, $v_{\theta}$ is only defined at anchor timesteps. In contrast, $v_{\phi_k}$ is defined at all timesteps. Since the expectation is taken over $t$, this mismatch makes the definition of $\Delta_k$ unclear.

(iii) My main theoretical concern is with Proposition A.7. As far as I understand, the proof does not include any updates related to IDA, since there are no updates over $k$ shown in the argument. Despite this, the authors make conclusions about the behavior of the procedure. It is possible that I may be missing some detail, but this seems to be mainly due to unclear definitions discussed earlier.

(iv) The proposed Intra-Segment Guidance (ISG) loss (Equation 12) appears closely related to the loss function used in Consistency Trajectory Models (CTM) [1]. If we set $s = \tau_{i-1}, u = t_{\text{mid}}, \tau_i = t$, ISG becomes a specific case of the CTM loss. If this connection is correct, the novelty of ISG should be further clarified and better distinguished from prior work.

(v) Equation 4 seems to contain a conceptual error. At $t = 0$, the forward process should converge to a delta distribution centered at $X_0$, yet the formula implies convergence to a standard Gaussian. This contradicts the standard behavior of the forward process. Moreover, in the appendix, the authors denote $X_1$ as the data variable, which adds further confusion regarding the consistency of notation.

[1] https://arxiv.org/abs/2310.02279 Consistency Trajectory Models: Learning Probability Flow ODE Trajectory of Diffusion

**Questions:**

(i) In Figure 3, I noticed that TTUR(10) appears to result in more unstable behavior compared to TTUR(5). Intuitively, a larger update ratio should allow for better optimization of the inner problem, potentially leading to more stable training. Maybe authors have some intuitions about that and can provide  an intuitive explanation for this behavior?

(ii) There seems to be a typo in line 726: "Eq. equation". Please correct it.

(iii) In the proof of Lemma A.3, could the authors explain why the inequality $\mathbb{E}||v_{\phi_{k+1}} - v_{\theta_k}|| \leq L ||\phi_{k+1} - \theta_k||$ holds under Assumptions A.1?

(iv) While I recognize that most reported metrics reflect human preferences to some extent, human evaluation remains an important standard in large-scale generative model assessments. Could the authors clarify why no human evaluation was included, and whether such comparisons might be provided in future work?

---

> ### Author Response · Authors · 2025-11-21
> **Response to Reviewer iTJr (Part I)**
>
> ### W5: Clarification on Eq. (4) and the roles of $X_0$/$X_1$.
>
> We thank the reviewer for pointing this out. Eq. (4) indeed contains a typo. Our intended forward process in the FM-OT example is the standard linear Gaussian path that perturbs data $X_0$ to Gaussian noise,
> $$
> X\_t = (1-t) X\_0 + t \xi,\qquad \xi \sim \mathcal N(0,I),
> $$
> which implies the conditional distribution
> $$
> q(X_t \mid X_0) = \mathcal N\big((1-t)X_0,\; t^2 I\big).
> $$
> This satisfies the desired boundary conditions: at $t=0$ the forward process reduces to a delta distribution at $X_0$ (clean data), and at $t=1$ we obtain $X_1 = \xi \sim \mathcal N(0,I)$ (standard Gaussian noise). In the current draft we mistakenly wrote $q(X_t \mid X_0) = \mathcal N(tX_0,(1-t^2)I)$, which corresponds to the opposite direction (noise $\to$ data) and conflicts with the textual description. We will correct Eq.~(4) in the revised version.
>
> In addition, as the reviewer correctly observes, in the current appendix, line 675 incorrectly states “$X_0 \sim \mathcal N(0,I), X_1 = G_\theta(\varepsilon)$”, reusing $X_0$ and $X_1$ with the opposite roles. In the revision we will remove this conflicting sentence and instead define the generator-parameterized target field directly in terms of the FM path $(X_0,X_t,X_1)$, i.e.,
> $$
> \hat v_\theta(x,t) := \mathbb{E}[X_1 - X_0 \mid X_t = x],
> $$
> consistent with the notation in Section 2.
>
> We also stress that this FM-OT example in Eq. (4) is only used for intuition in the preliminaries; all theoretical results for IDA in Appendix A rely only on the generic forward kernel $q(X\_t \mid X\_0)$ and the induced marginals $p\_g(X\_t)$, not on the particular parameterization of Eq. (4). Therefore this typo and the notational inconsistency do not affect our proofs or conclusions.
>
> ### W1: Definition of $\hat v_\theta(X_t,t)$ and randomness of $X_1$.
>
> We appreciate the reviewer’s careful reading and agree that our original notation around $\hat v_\theta$ was ambiguous. In particular, as pointed out, writing
> $$
> \hat v_\theta(X_t,t) = X_1 - X_0
> $$
> can be misread as if we were treating a conditional quantity as deterministic “given $X_1$”, whereas $X_0, X_t, X_1$ are indeed random variables along the forward FM path.
>
> In the revised version, we will make the following clarification. Under the unified notation described in our response to W5, let $(X_0, X_t, X_1)$ denote the random path induced by the forward kernel $q(X_t \mid X_0)$, where $X_0$ denotes a data sample and $X_1$ denotes the Gaussian noise endpoint. Thus $X_0, X_t, X_1$ are all random variables. We then define the generator-induced FM target field as the conditional expectation
> $$
> \hat v_\theta(x,t) := \mathbb{E}[X_1 - X_0 \mid X_t = x],
> $$
> which is a deterministic measurable function of $(x,t)$.
>
> All error terms in Appendix A (e.g.,
> $
> \bar d\_k = \mathbb{E}\_{t,X\_t}\big\|\big\|v_{\phi_k}(X_t,t) - \hat v_{\theta_k}(X_t,t)\big\|\big\|
> $  )
> are taken with respect to the joint law of $(t,X_t)$ and only involve this deterministic field $\hat v_{\theta_k}(\cdot,\cdot)$. In practice, we can approximate this conditional expectation by the per-sample path target $X_1 - X_0$ along each FM path.
>
> We have updated the text in Appendix A.1 to explicitly introduce $\hat v_\theta(x,t) := \mathbb{E}[X_1 - X_0 \mid X_t = x]$ and to clarify that $X_0$ and $X_1$ are random variables along the forward path, while $\hat v_\theta$ is a deterministic generator-induced FM target field evaluated at random $X_t$.

---

> > ### Author Response · Authors · 2025-11-21
> > **Response to Reviewer iTJr (Part II)**
> >
> > ### W2: Definition of $\Delta_k$ and the domain of $v_{\theta_k}$.
> >
> > We thank the reviewer for raising this point. Our intention was that both $v_{\phi_k}$ and $v_{\theta_k}$ are defined as velocity fields over the entire time interval $t \in [0,1]$, and that $\Delta_k$ measures their discrepancy on this common domain. The generator $G_\theta$ takes $(x,t)$ as input via a time embedding, so it induces a velocity field
> > $$
> > v_\theta(x,t) \in \mathbb{R}^d, \qquad \forall\, t \in [0,1],
> > $$
> > however, during DMD training, it only receives gradient updates at a sparse set of anchor timesteps (e.g., 4 fixed $t$’s). Our note on line 670 (“the generator is trained on a sparse set of timesteps”) was meant to emphasize where gradients are applied, not to suggest that $v_{\theta_k}$ is undefined at other $t$.
> >
> > Formally, in Appendix A.2 we consider the field gap
> > $$
> > \Delta_k :=\; \mathbb{E}\_{t,X_t}\big\|\big\|v_{\phi_k}(X_t,t) - v_{\theta_k}(X_t,t)\big\|\big\|,
> > $$
> > where the expectation is taken over the same time distribution as in our FM path construction. Since both $v_{\phi_k}$ and $v_{\theta_k}$ are well-defined functions of $(x,t)$ for all $t\in[0,1]$, this quantity is well-defined and serves as an aggregate measure of the discrepancy between the fake model’s velocity field and the generator-parameterized field along the time dimension.
> >
> > In the revised version, we have explicitly stated in Appendix A that $v_{\theta}(x,t)$ is defined for all $t\in[0,1]$ through the time embedding of $G_\theta$, and rephrase the Note on line 670 as “the generator only receives gradient updates at these anchor timesteps” to avoid the impression that $v_{\theta_k}$ is only defined at anchor times. We will also update the definition of $\Delta_k$ in Appendix A.2 to the explicit form above.

---

> > > ### Author Response · Authors · 2025-11-21
> > > **Response to Reviewer iTJr (Part III)**
> > >
> > > ### W3: Role of IDA in Proposition A.7.
> > >
> > > We appreciate the reviewer’s concern about how the IDA update enters Proposition A.7 and apologize for the confusion caused by our presentation. Below we clarify the role of IDA in our analysis and how it interacts with Proposition A.7.
> > >
> > > Our analysis in Appendix A can be separated into two parts:
> > >
> > > - applying IDA to control the tracking errors $(e_k,\bar d_k)$
> > > - a generic conversion from field error to KL gap and an $\varepsilon$-best-response bound.
> > >
> > > (i) IDA-specific part (Section A.2).
> > > Under the IDA update
> > > $$
> > > \phi_{k+1} = \lambda \phi_k + (1-\lambda)\theta_{k+1} \quad \text{(Eq.~9)},
> > > $$
> > >
> > > together with Assumption A.1, Lemma A.3 and Lemma A.4 establish coupled recursions for the parameter tracking error $e_k := \|\phi_k-\theta_k\|$ and the field tracking error $\bar d_k$. Intuitively, the vanilla DMD min–max updates do not control how far the fake parameters $\phi_k$ can drift away from the generator parameters $\theta_k$, especially for large scale models. The IDA step explicitly couples $\phi_{k+1}$ to $\theta_{k+1}$ via a proximal averaging, yielding a recursion of the form
> > > $$
> > > e_{k+1} \le \lambda e_k + \lambda\|\|\theta_{k+1} - \theta_k\|\|,
> > > $$
> > > which in turn implies that the field tracking error remains bounded:
> > > $$
> > > \limsup_{k\to\infty} \bar d_k \le \bar\beta_\star < \infty
> > > \quad\text{(Proposition~A.5).}
> > > $$
> > > Thus, the dependence on the IDA update is already encoded in the boundedness of $(e_k,\bar d_k)$.
> > >
> > > (ii) “Field error to KL gap” part (Section A.3 and Proposition A.7).
> > > Section A.3 then relates the velocity-field error to the pathwise KL via Assumption A.6 (Fisher-type control and score–velocity relation), obtaining
> > > $$
> > > \mathbb{E}\_t D_{\mathrm{KL}}\big(p\_g(X\_t) \Vert p_f(X\_t)\big)
> > >    \le C \mathbb{E}\big\|\big\|v_{\phi_k}(X_t,t) - v_{\theta_k}(X_t,t)\big\|\big\|_2^2.
> > > $$
> > >
> > > Proposition A.7 is a generic consequence of this inequality: for each fixed iterate $k$, it rewrites the right-hand side in terms of the $L_2$ versions of the tracking and self-consistent errors $(\tilde d_k,\tilde\beta_k)$ and defines
> > > $$
> > > \varepsilon\_k := 2C\big(\tilde d\_k\^2 + \tilde\beta\_k\^2\big)
> > > $$
> > > so that
> > > $$
> > > \mathbb{E}\_t D\_{\mathrm{KL}}\big(p_g(X_t) \Vert p_f(X_t)\big) \le \varepsilon_k.
> > > $$
> > > This step does not introduce new updates over $k$; it converts the field error into a KL gap bound.
> > >
> > > (iii) Combining the two (Section A.4).
> > > IDA enters the final $\varepsilon$-best-response statement by combining the above two ingredients. Under IDA, Proposition A.5 guarantees that $(\tilde d_k,\tilde\beta_k)$ remain bounded, hence $\sup_k \varepsilon_k \le \varepsilon < \infty$ for some finite $\varepsilon$. Plugging this bound into the decomposition of $V(\theta,\phi)$ in Proposition A.2 yields Eq. (27), namely that the inner loop delivers an $\varepsilon$-best response in expectation.
> > >
> > > We agree that our original naming of Proposition A.7 (“IDA maintains an $\varepsilon$-best response”) obscures this separation between (i) the IDA-specific control of $(e_k,\bar d_k)$ in Section A.2 and (ii) the general field-to-KL conversion in Section A.3. In the revised version, we have renamed Proposition A.7 to emphasize that it is a “field error $\Rightarrow$ $\varepsilon$-best-response bound”, and made Section A.4 explicitly state that combining Proposition A.5 (which uses the IDA update in Eq. 9) with Proposition A.7 yields the claimed $\varepsilon$-best-response property for IDA.

---

> > > ### Comment · Reviewer_iTJr · 2025-11-26
> > >
> > > I would like to thank the authors for their responses. Upon revising the updated text, I find that most of my concerns regarding the practical aspects of the work have been satisfactorily addressed. However, I still have some concerning about theoretical components.
> > >
> > > In particular, I remain unclear about the definition and interpretation of the quantity $v_{\\theta}(x, t)$. In the revised version, the authors state that this term "induces a velocity field from the generator." However, I find this formulation ambiguous, as the generator does not explicitly define a velocity field.
> > >
> > > To clarify, under the authors’ notation, if we consider a single-step generator, the model distribution is given by $p_g = G_{\\theta} \\text{\\#} p_{\\epsilon}$, where $p_{\\epsilon} = \\mathcal{N}(0, I)$. This approach can be naturally extended to multiple steps. For example, in a four-step generator, one alternates between applying the generator and adding Gaussian noise, repeating this process until reaching the initial distribution. Although the number of steps increases, the overall structure of the process remains the same. In this setup, no velocity field is explicitly defined. For this reason, I find the authors’ reference to a “velocity field induced by the generator” unclear and would appreciate a more precise definition.
> > >
> > > Given this formulation, I am seeking a precise and formal definition of $v_{\\theta}(x, t)$, rather than a heuristic explanation such as “induces a velocity field,” which remains insufficiently rigorous. The generator is defined as a discrete composition of mappings and does not, in this form, define a continuous-time velocity field. A concrete expression for this term is essential for ensuring the soundness of the theoretical analysis.
> > >
> > > I acknowledge that the main strengths of the paper lie in its empirical performance, which is indeed impressive. Therefore, I am inclined to raise my score. However, since IDA is one of the most important part of the paper, I want to be sure that corresponding proofs and theoretical analysis are correct. Once this clarification is provided, I will revisit the proofs in the revised text, and if the theoretical concerns are adequately resolved, I will be happy to increase my evaluation accordingly.

---

> ### Author Response · Authors · 2025-11-21
> **Response to Reviewer iTJr (Part IV)**
>
> ### W4: Relation to CTM and other flow–map based methods.
>
> We thank the reviewer for pointing out the connection to CTM. Conceptually, CTM and recent flow-map works [1,2,3] such as “Flow map matching with stochastic interpolants” and “Mean Flows for One-step Generative Modeling” all aim to learn a global flow operator $G_\theta(x_t,t,s)$ that maps a state at any time $t$ to a state at any other time $s$, i.e., an “any-to-any” trajectory model approximating the underlying PF-ODE dynamics. CTM instantiates this idea by distilling a trajectory network $G_\theta(x_t,t,s)$ to match the output of a multi-step ODE solver for arbitrary $(t,s)$, and further regularizes training via soft consistency using a student-EMA projector back to time $0$ together with an additional denoising score matching loss.
>
> In contrast, our Intra-Segment Guidance (ISG) is not designed to learn such a universal flow map or to replace the ODE solver. We work in a DMD-style few-step distillation setting with a fixed coarse timestep set $\{\tau_i\}$, and ISG acts as a lightweight segment-local guidance term that aggregates information within each interval $(\tau_{i-1},\tau_i]$ onto the chosen anchor $\tau_i$ to stabilize and improve a given few-step sampler. We do not match the student to a multi-step solver over arbitrary $(t,s)$, do not project both paths back to time $0$, and do not introduce a separate DSM loss as in CTM. Thus, while CTM/flow-map/MeanFlow methods and ISG all exploit consistency between different trajectories, they target different problems: the former build a general any-to-any trajectory/flow operator, whereas ISG is a simple local regularizer tailored to fixed-timestep few-step distillation. We will clarify this distinction and explicitly cite and discuss CTM as well as the above flow-map/MeanFlow works in the revised version.
>
> [1] Flow map matching with stochastic interpolants: A mathematical framework for consistency models
>
> [2] Mean Flows for One-step Generative Modeling
>
> [3] Align Your Flow: Scaling Continuous-Time Flow Map Distillation
>
> ### Q1: On why TTUR(10) is less stable than TTUR(5).
>
> We agree that, in an ideal bi-level optimization setting, a larger TTUR could in principle help the inner (fake) model get closer to its optimum and thus improve stability. In our DMD setting, however, the inner problem is only solved by a few noisy SGD steps on top of a highly non-stationary generator distribution $p_g$, especially at early stages when $p_g$ is still very far from the teacher.
>
> Intuitively, there are two competing effects when we increase TTUR. On the one hand, more fake updates per generator step can improve the inner approximation. On the other hand, in the early phase the generator produces low-quality and rapidly drifting samples; giving the fake model too many updates on this transient, biased $p_g$ makes it strongly adapt to a poor distribution and behave like an overly strong discriminator. At the same time, for a fixed training budget, a larger TTUR means that the generator itself is updated less frequently. Thus a moderate ratio such as TTUR(10) can be the most problematic: the fake model has enough steps to overfit the wrong distribution, while the generator does not have enough steps to correct itself, leading to the large spikes observed in Fig. 3.
>
> For a very large ratio such as TTUR(20), the generator changes so slowly that $p_g$ becomes locally closer to stationary during the inner updates, so the “better inner approximation” effect starts to partly compensate for the overfitting effect. This explains why TTUR(20) is slightly more stable than TTUR(10), although it is still far from satisfactory and much worse than our IDA-based training. Overall, in our highly non-stationary bi-level problem, the dependence on TTUR is non-monotonic: increasing TTUR does not guarantee more stability, which motivates controlling the fake–generator gap via IDA instead of simply using larger TTUR ratios.
>
> ### Q2: Typo in line 726.
>
> Thank you for catching this. We have fixed the typo “Eq. equation” in line 726 in the revised paper.

---

> ### Author Response · Authors · 2025-11-21
> **Response to Reviewer iTJr (Part V)**
>
> ### Q3: Regarding Lemma A.3 (param→field Lipschitz).
>
> We thank the reviewer for pointing out this ambiguity. Our intent in Assumption~A.1 was to assume that the velocity field is Lipschitz with respect to the network parameters, for any pair of parameter vectors in a neighborhood of the training trajectory. Formally, let $\omega$ denote a generic parameter vector (covering both the fake and generator networks, which share the same architecture), and define $v_\omega(x,t)$ as its induced velocity field. We assume that there exists $L > 0$ such that for all $\omega,\omega'$ in a common neighborhood $\mathcal U \subset \mathbb{R}^d$,
>
> $$
> \mathbb{E}\_{t,X\_t\sim p\_g}\big\|\big\|v\_{\omega'}(X\_t,t) - v_{\omega}(X\_t,t)\big\|\big\|
> \le L\|\|\omega' - \omega\|\|.
> $$
>
> Under this formulation, the inequality used in Lemma~A.3 follows directly by taking $\omega' = \phi_{k+1}$ and $\omega = \theta_k$:
> $$
> \mathbb{E}\_{t,X\_t}\big\|\big\|v\_{\phi\_{k+1}}(X_t,t) - v_{\theta\_k}(X\_t,t)\big\|\big\|
> \le L\|\|\phi_{k+1} - \theta_k\|\|.
> $$
>
> In practice, the fake and generator networks share the same architecture and are initialized identically ($\phi_0=\theta_0$). They are updated with small gradient steps, and we further apply the IDA update $\phi_{k+1} = \lambda\phi_k + (1-\lambda)\theta_{k+1}$, so their parameter trajectories $(\phi_k,\theta_k)_k$ remain not far from each other and stay within a common neighborhood $\mathcal U$. Our Lipschitz assumption is only required on this local neighborhood, which makes it a mild regularity condition.
>
> In the revised version, we have updated Assumption A.1 to state this Lipschitz property for arbitrary parameter pairs $(\omega,\omega')$ in a local neighborhood (rather than only for consecutive fake parameters $(\phi_{k+1},\phi_k)$), and explicitly mention that it applies to both the fake and generator parameters.
>
> ### Q4: Human Evaluation.
>
> We agree that human evaluation is an important standard for assessing large-scale generative models. Following the reviewer’s suggestion, we have conducted a human preference study and will include the results in the revised paper.
>
> We recruited about 20 participants and evaluated SenseFlow across three model families: SDXL (vs. DMD2 and Hyper-SD), SD 3.5 (vs. SD 3.5 Large Turbo and SD 3.5 Large), and FLUX (vs. Hyper-FLUX and FLUX-Turbo-Alpha). We used $\sim$80 diverse prompts (from prior distillation works plus GPT-generated prompts). For each prompt and model family, participants saw three anonymized images (SenseFlow and two baselines) in random order and ranked them by overall preference, image quality, and text–image alignment.
>
> From these rankings we derived pairwise preference rates (detailed in Fig. 14 in the Appendix). SenseFlow is preferred in roughly 60–72% of comparisons on SDXL, 58–68% on SD 3.5 Large, and 64–72% on FLUX across the three criteria, consistently outperforming strong baselines. These results corroborate our reported metrics in the main paper, and we plan to further scale up the human evaluation in future.
>
> Besides, in the revised paper we also included more visualization comparison results between our method and strong baselines on SDXL, SD 3.5 Large and FLUX.1 dev, shown in Fig. 15-17. At the same NFE, our method consistently produce images with sharper local details, improved aesthetics, and higher perceptual realism compared to baselines such as DMD2-SDXL, SD 3.5-Large-Turbo, and Hyper-FLUX, etc. which aligns with the higher human-preference performance.

---

> ### Author Response · Authors · 2025-11-27
> **Response to Reviewer iTJr**
>
> We thank the reviewer for the positive follow-up and are glad that most practical concerns have been resolved. We also appreciate the careful scrutiny of the theoretical components—below we provide a precise, formal definition of $v\_\theta(x,t)$ (and related quantities), and we have revised Appendix A accordingly to remove the ambiguity of the previous “induces a velocity field” phrasing.
>
> **1) What exactly is $v\_\theta(x,t)$? (formal definition)**
>
> In our setup, the *student generator* shares the same modeling form as the flow-matching teacher: it is a **time-conditional velocity network** (e.g., DiT-style backbone) that takes $(x,t)$ as input and outputs a velocity prediction at that time. Concretely, we define
> $$
> v_\theta(x,t) := f_\theta(x,t), \quad t\in[0,1],
> $$
> where $f_\theta$ is the generator’s neural network. This makes $v_\theta(\cdot,\cdot)$ a deterministic function (given $\theta$ defined for all $t\in[0,1]$. To avoid confusion, in the revised Appendix A we also explicitly note that we may write $v_\theta(X_t,t)$ when evaluating this deterministic field at a input $(X_t,t)$.
>
> **2) Why a “$K$-step generator” does not contradict the definition above**
>
> “$K$-step generator” refers to **how we *use* the $v_\theta$ at inference/training**, namely by numerically integrating an ODE along a chosen timestep schedule with $K$ function evaluations (e.g., Euler steps). In other words, $K$ controls the discretization of the solver, not whether $v_\theta(x,t)$ exists. Even if DMD training supervision is applied only at a sparse set of anchor timesteps, the network $f_\theta(x,t)$ is still defined for every $t\in[0,1]$; anchors only specify where gradients are evaluated during optimization.
>
> **3) Generator-induced FM target field $\hat v_\theta(x,t)$ and the inner-loop objective (tracking)**
>
> To connect the inner-loop objective to flow-matching targets, we define the *generator-induced FM target field* along the FM path (under the forward kernel $q(X_t\mid X_0)$ with $X_0\sim p_g$, $X_1\sim\mathcal N(0,I)$):
> $$
> \hat v_\theta(x,t) := \mathbb E\left[X_1 - X_0 \middle| X_t=x\right].
> $$
> Using this, we define the **tracking error** (the inner-loop optimization target):
> $$
> \bar d_k := \mathbb E_{t,X_t}||v_{\phi_k}(X_t,t)-\hat v_{\theta_k}(X_t,t)||,
> $$
> so smaller $\bar d_k$ means the fake model $v_{\phi_k}$ (and thus $p_f$) better tracks the generator-induced path distribution $p_g$. This is now stated explicitly and earlier in Appendix A to clarify the role of $\bar d_k$ as the inner-loop objective.
>
> Note: In practice, when generator is deployed with a 4-step solver, $p_g$ corresponds to the distribution induced by the resulting 4-step samples.
>
> **4) Self-consistent error and why it is needed in the analysis**
>
> We define the **self-consistent error** as
> $$
> \bar\beta_k := \mathbb E_{t,X_t}||v_{\theta_k}(X_t,t)-\hat v_{\theta_k}(X_t,t)||.
> $$
> This term measures how well the generator’s own velocity prediction agrees with its pathwise FM target along the trajectory. Importantly, we do **not** require $\bar\beta_k\to 0$; our theory assumes $\sup_k \bar\beta_k < \infty$, which is also the condition used in Proposition A.5/A.7 to ensure the inner-loop tracking remains controlled and to obtain the $\varepsilon$-best-response bound.
>
> **5) Empirical support via step consistency (Appendix A.5)**
> To support the assumed boundedness of $\bar\beta_k$, we provide empirical evidence of **step consistency**: a student trained with 4-step supervision produces samples that remain visually very close when re-evaluated with finer Euler discretizations (8/16 steps) under fixed prompts/seeds, both without IDA and with IDA (where quality further improves). Intuitively, if the learned field were highly inconsistent between anchor timesteps, refining the discretization (e.g., $4\to 16$) would typically induce noticeable trajectory drift and different endpoints; this is not observed in our results. We therefore use step consistency as an empirical sanity check that the learned time-conditional field does not exhibit unstable behavior between anchors, consistent with $\sup_k \bar\beta_k<\infty$.
>
> **Summary of changes in the revised Appendix A**
>
> * Replaced the ambiguous “induces a velocity field” phrasing with the explicit definition $v_\theta(x,t):=f_\theta(x,t)$.
> * Reorganized definitions to introduce (i) $(p_g,p_f,p_r)$, then (ii) the generator-induced FM target $\hat v_\theta$ and tracking objective $\bar d_k$ (inner loop), and then (iii) the self-consistent error $\bar\beta_k$.
> * Strengthened Appendix A.5 to explain how step consistency provides empirical support for bounded self-consistent error, which is the assumption used by the $\varepsilon$-best-response analysis.
>
> We hope these clarifications address the reviewer’s concern about rigor and make the theoretical components fully precise and self-contained.

---

### Official Review · Reviewer_j3tb · 2025-10-29

**Soundness:** 3
**Presentation:** 4
**Contribution:** 3
**Rating:** 6
**Confidence:** 5

**Summary:**

This paper addresses the limitations of Distribution Matching Distillation (DMD and DMD2)—a state-of-the-art timestep distillation framework for diffusion models—when applied to large-scale flow-based text-to-image models such as SD3.5 and FLUX. The authors first analyze the performance of vanilla DMD on these large models and identify key challenges in scaling. To overcome them, they propose three improvements: Implicit Distribution Alignment (IDA), which enforces consistency between the weights of the fake model and the student model; Intra-Segment Guidance (ISG), which redistributes timestep denoising importance learned from the teacher model; and a VFM-based discriminator, which leverages the rich prior knowledge encoded in large foundation models.

**Strengths:**

1. The paper is well-written and well-structured, presenting a clear and compelling motivation for the study.
2. The proposed approach is thoughtfully designed, addressing the common challenges of stability and scalability in distribution-matching distillation.
3. The experiments are comprehensive, supported by detailed analysis and clear explanations.

**Weaknesses:**

1. The effect of ISG is not very clear. Although Table 2 shows slight improvements in quality and human-preference metrics, no qualitative comparisons are provided for this component. Including an additional visualization—similar to Figure 3—or a reconstruction loss curve could better illustrate ISG’s impact on training stability and convergence speed.
2. The design of the VFM discriminator also requires further clarification, particularly regarding the rationale for incorporating another reference image feature as a conditioning input. Moreover, additional visual comparisons would strengthen the paper, especially since you claim a trade-off between distribution coverage (FID-T) and human-preference scores when using this discriminator.

**Questions:**

1. If possible, please provide additional visualizations illustrating the effect of ISG on the training process.
2. The use of strong foundation models such as DINOv2 and CLIP for the discriminator is reasonable. However, the incorporation of the reference image feature as an additional conditioning input deviates from the vanilla DMD design. Could the authors clarify the motivation for this choice and discuss its impact on image quality and diversity?
3. How many images were used during training?
4. In Table 1, it is strongly recommended to include DMD2 results for SD3.5 and FLUX to demonstrate the proposed method’s effectiveness over DMD2 when applied to flow-based models, as claimed.
5. Table 2 is not comprehensive and should include more ablation variants that isolate or combine each proposed component.

---

> ### Author Response · Authors · 2025-11-21
> **Response to Reviewer j3tb (Part I)**
>
> ### W1&Q1: ISG’s impact on training stability and convergence speed.
>
> We thank the reviewer for pointing out that the impact of ISG was not sufficiently illustrated. In the revised version, we add a training-dynamics visualization for SD 3.5 Large (in Fig. 12 in the revised paper), where we plot FID and FID-T on COCO-5k as a function of training iterations for our full method and an ablation without ISG. The corresponding FID-T values are also summarized in table below.
>
> | **Method**    | **1.5k**  | **3k**    | **4.5k**  | **6k**    | **9k**    | **12k**   |
> | ------------- | --------- | --------- | --------- | --------- | --------- | --------- |
> | Ours (w/ ISG) | **14.48** | **22.32** | **17.65** | **18.11** | **16.18** | **15.20** |
> | Ours w/o ISG  | 138.2     | 24.99     | 19.43     | 19.12     | 18.07     | 19.06     |
>
> As shown above, ISG mainly improves the stability and convergence speed of training. At 1.5k iterations, the model without ISG
> still exhibits extremely poor alignment to the teacher (FID-T $\approx$138), whereas our full method with ISG has already entered a
> reasonable regime (FID-T $\approx$14.48). As training proceeds, both variants become more stable, but the ISG model consistently attains lower FID-T throughout training (e.g., 16.18 vs. 18.07 at 9k iterations, and 15.20 vs. 19.06 at 12k iterations). A similar gap is observed on the FID curves in Fig. 12. These results support our claim that ISG helps redistribute timestep importance within each segment, making the student updates less sensitive to under-trained timesteps and leading to faster and more stable convergence.
>
>
>
> ---
>
> ### W2&Q2: Design and motivation of the VFM discriminator.
>
> We thank the reviewer for asking for more clarification on the VFM-based discriminator. Figure 9 in the paper visualizes the design: besides the generated image $x_g$, the discriminator also receives a reference image $x_{\text{ref}}$ and the text prompt.
>
> **Motivation.** The reference image is used as a *conditional anchor* for the discriminator. In practice, $x_{\text{ref}}$ is a high-quality, semantically aligned sample from the ground-truth data, while $x_g$ comes from the student (as fake samples). Conditioning on $(x_{\text{ref}}, \text{text})$ allows the discriminator to focus on the *relative* discrepancy between fake and real samples for the same semantic content, rather than deciding “good vs. bad’’ in isolation.
>
> **Ablation on reference conditioning.** To quantify the effect of this design, we train two SD 3.5 Large students under the same setting: one with the full VFM discriminator and one where the reference branch is removed (the discriminator only sees $x_g$ and the text prompt). After 9k iterations, the full model consistently outperforms the variant without reference conditioning on all metrics, as summarized in
> the table below. In particular, FID-T improves from **17.77** to **14.76**, and human-preference metrics such as HPSv2, AESv2, PickScore, and ImageReward all increase (e.g., ImageReward from **1.030** to **1.103**).
>
> | **Method**    | **FID-T** $\downarrow$ | **CLIP** $\uparrow$ | **HPSv2** $\uparrow$ | **AESv2** $\uparrow$ | **PickScore** $\uparrow$ | **ImageReward** $\uparrow$ |
> | ------------- | ---------------------- | ------------------- | -------------------- | -------------------- | ------------------------ | -------------------------- |
> | Ours (w/ ref) | **14.76**              | **0.3274**          | **0.2850**           | **5.518**            | **22.86**                | **1.103**                  |
> | Ours w/o ref  | 17.77                  | 0.3259              | 0.2833               | 5.500                | 22.57                    | 1.030                      |
>
> **More visualization results between our method and existing baselines.** To better illustrate the claimed trade-off, we also add more visual comparisons between SenseFlow and existing baselines in the revised appendix B.11 for SDXL, SD 3.5 Large, and FLUX (Fig. 15–17). At the same NFE, our VFM-equipped students consistently produce images with sharper local details, improved aesthetics, and higher perceptual realism compared to baselines such as DMD2-SDXL, SD 3.5-Large-Turbo, and Hyper-FLUX, etc. which aligns with the higher human-preference scores in Tables 1–2.

---

> ### Author Response · Authors · 2025-11-21
> **Response to Reviewer j3tb (Part II)**
>
> ### Q3: Number of training images.
>
> We trained our models on a filtered subset of LAION-5B, where we keep images with an aesthetic score $\geq 5.0$ and a shorter side of at least 1024 pixels. This filtering yields roughly **500k** image–text pairs, from which we sample training batches.  In practice, we use 8 A100 GPUs with a global batch size of 8. A typical SDXL distillation run is trained for 18k iterations, which corresponds to about **144k** image–text pairs ($18\text{k}\times 8$). For SD 3.5 Large, we
> train for 27k iterations, i.e., about **216k** image–text pairs. We will clarify both the dataset size and these iteration/batch statistics in the revised Implementation Details section.
>
> ---
>
> ### Q4 & Q5: Compare with DMD2 & More ablation variants of Table 2.
>
> We appreciate the reviewer’s suggestions regarding (i) including DMD2 results for SD 3.5 Large and FLUX, and (ii) providing more systematic ablations for the proposed components.
>
> **On DMD2 for SD 3.5 Large and FLUX.** For SDXL, we directly report the official DMD2-SDXL model as a strong baseline in the main table. For SD 3.5 Large and FLUX, however, there is currently no public DMD2 implementation. We therefore implemented a faithful DMD2-style latent discriminator on top of the DiT backbone ourselves and attempted to train “DMD2-SD 3.5 Large’’ and “DMD2-FLUX’’ under the same settings. Despite extensive tuning (e.g., increasing the TTUR up to 20), these models failed to converge in practice: SD 3.5 Large collapses to nearly all-black samples. Only after replacing the vanilla discriminator with our VFM-based discriminator did training become to produce meaningful images, but still unstable; subsequent addition of IDA and ISG then gradually brought the performance to the level reported for SenseFlow. That's the reason why we do not include these failed DMD2 variants in Table 1. Instead, the revised paper includes qualitative examples of this progression in Fig. 13: from DMD2 (black images) to DMD2+VFM, then DMD2+VFM+IDA (Ours w/o ISG), Ours (full), and DMD2+VFM+ISG (Ours w/o IDA). This visual sequence directly illustrates why plain DMD2 is difficult to scale to SD 3.5 Large / FLUX and how our modifications remedy the failure modes.
>
> **More comprehensive ablations.**  Further, we add an expanded ablation table below (also Tab. 9 in the revision) together with the above visual comparison. For SD 3.5 Large, the table includes:
> 	(a) our full SenseFlow model,
> 	(b)  w/o ISG,
> 	(c)  w/o IDA, and
> 	(d) w/o ISG \& w/o IDA.
>
> Performance degrades along this path, showing that both IDA and ISG contribute substantially and that removing both leads to a clear collapse. For SDXL, the same table reports DMD2-SDXL, DMD2 w VFM, and Ours-SDXL, which isolates the effect of the VFM discriminator (DMD2 vs. DMD2 w VFM) and then the additional gains from ISG on top of it (DMD2 w VFM vs. Ours-SDXL).
>
> Together, the table below and the accompanying qualitative grid (Fig. 13 in the revised paper) provide a step-by-step view of how the three components—VFM discriminator, IDA, and ISG—each improve the model and how their combination yields the strongest overall performance, while also explaining why a direct DMD2 baseline is not meaningful for SD 3.5 Large and FLUX.
>
> **Table: Ablation Study Results of IDA, ISG, and VFM Discriminator.**
>
> | **Method**       | **FID-T** $\downarrow$ | **HPSv2** $\uparrow$ | **Pick** $\uparrow$ | **ImageReward** $\uparrow$ | **AESv2** $\uparrow$ |
> | ---------------- | ---------------------- | -------------------- | ------------------- | -------------------------- | -------------------- |
> | **SD 3.5 Large** |                        |                      |                     |                            |                      |
> | Ours             | **13.38**              | **0.3015**           | **23.03**           | **1.1713**                 | **5.482**            |
> | w/o ISG          | *17.00*                | *0.2971*             | *22.75*             | *1.0186*                   | *5.453*              |
> | w/o IDA          | 17.83                  | 0.2800               | 22.47               | 0.9365                     | 5.407                |
> | w/o ISG, w/o IDA | 43.84                  | 0.2555               | 20.60               | 0.3828                     | 5.102                |
> | **SDXL**         |                        |                      |                     |                            |                      |
> | DMD2-SDXL        | **15.04**              | 0.2964               | 22.98               | 0.9324                     | 5.530                |
> | DMD2 w VFM       | 18.55                  | *0.2995*             | *23.00*             | *0.9744*                   | *5.625*              |
> | Ours-SDXL        | *17.76*                | **0.3010**           | **23.17**           | **0.9951**                 | **5.703**            |

---

> > ### Author Response · Authors · 2025-11-27
> > **Comment by authors**
> >
> > Thank you for handling our manuscript and providing valuable feedback. We hope that our responses have sufficiently addressed the concerns you raised. We welcome more discussion if you have more questions and suggestions. As the discussion deadline is approaching, we would be very grateful if you could take a moment to review our reply.

---

> > > ### Comment · Reviewer_j3tb · 2025-11-28
> > >
> > > Thanks for your response to clarify the contributions of ISG and VFM discriminator. It addresses my concerns. I keep my original positive score.

---

### Official Review · Reviewer_xdRf · 2025-10-30

**Soundness:** 3
**Presentation:** 3
**Contribution:** 2
**Rating:** 6
**Confidence:** 3

**Summary:**

The paper addresses instability in Distribution Matching Distillation (DMD/DMD2) when scaling to large flow-based text-to-image (T2I) teachers (e.g., SD-3.5 Large, FLUX.1-dev), identifying poor fake-network tracking and sparse timestep supervision as key issues. It introduces $\textbf{Implicit Distribution Alignment (IDA)}$, which stabilizes training by progressively aligning the fake network with the generator ($\phi \leftarrow \lambda \phi + (1-\lambda)\theta$), and $\textbf{Intra-Segment Guidance (ISG)}$, which samples intermediate denoising steps to better cover under-trained timesteps. A VFM-based discriminator (frozen DINOv2 + CLIP) provides semantic guidance. On SDXL, SD-3.5 Large, and FLUX.1-dev, SenseFlow trains 4-step generators that nearly match 80-step teacher quality (e.g., GenEval 0.7098 vs. 0.7140) with minimal overhead ($\approx$3--6\% for ISG, 0.6--4\% for IDA).

**Strengths:**

+ IDA directly addresses the brittle inner loop in DMD via an explicit, cheap parameter interpolation; Appendix A formalizes that this bounds the generator–fake field gap. The “Training‑hours vs. FID” plot (Fig. 3) shows markedly smoother convergence with IDA on SD‑3.5 Large.

+ Experiments span three strong teachers (SDXL 2.6B, SD‑3.5 Large 8B, FLUX.1‑dev 12B) with contemporary baselines (LCM/PCM/Lightning/Hyper, SD‑3.5‑Turbo, FLUX‑Turbo/‑schnell). Results cover COCO‑5K, GenEval, and T2I‑CompBench; for SD‑3.5 the 4‑step Euler variant is best or second‑best across metrics and best in 5/6 CompBench categories.

+ Tables/figures ablate the contributions of IDA, ISG, and VFM‑disc (e.g., removing both tanks SD‑3.5 quality from FID‑T 13.38 → 43.84), and they quantify overheads (ISG ≈3–6%; IDA ≈0.6–4%). Appendix A.5’s step‑consistency visuals support the assumption behind IDA.

**Weaknesses:**

+ The paper notes that adding the VFM discriminator improves human‑preference proxies while slightly raising FID‑T (interpreted as reduced diversity). However, there is no explicit diversity study (e.g., user study, precision–recall curves over seeds). Given the important of this trade‑off in distillation, a quantitative diversity analysis would strengthen the claims.

+ The authors seem to omit IDA on smaller models SDXL but claims generality. A small ablation whether IDA helps small-scale models or not can clarify further the importance of IDA. Timestep settings differ per model (Appendix B.1) but the impact of these choices isn’t studied.

+ All the results are 4‑step; but the prposed method should be able to handle 1-2 step(s) sampling. Showing at least 2‑step results or failure modes would calibrate how far SenseFlow goes in the low‑NFE regime.

**Questions:**

+ Please add benchmarks about diversity as I mentioned above
+ Does IDA hurt/help in that small-scale models setting (SD1.5 and SDXL)? Also, can you discuss how anchor selection (Appendix B.1) affects results?
+ Can SenseFlow handle 1-2 step(s) sampling ? If yes then please show comparison and benchmarks

---

> ### Author Response · Authors · 2025-11-21
> **Response to Reviewer xdRf (Part I)**
>
> ### W1&Q1: Diversity study
>
> We thank the reviewer for pointing out that our discussion of the FID–human-preference trade-off (in Sec. 4.3) would benefit from an explicit diversity analysis, rather than stating the effect only qualitatively.
>
> **Setup.**
> Following the suggestion, we construct a prompt validation set and, for each prompt, generate 16 samples per model. On this set we compute two diversity measures across all image pairs: (i) *LPIPS-diversity*, defined as the average pairwise LPIPS distance between samples; and (ii) *CLIP-diversity*, where we feed all images through a pretrained CLIP image encoder, $\ell_2$-normalize the features, and take the average of $1-\text{cosine-similarity}$ over all pairs. Both metrics are averaged over prompts, and higher values indicate higher sample diversity.
>
> **Results.**
> The quantitative results are shown in the table below. For SDXL, Ours-SDXL achieves LPIPS-diversity **0.6002** vs. **0.5960** for the DMD2-SDXL baseline, and CLIP-diversity **0.0802** vs. **0.0985**.
> For SD 3.5 Large, our method obtains LPIPS-diversity **0.5664** vs. **0.5659** for the SD~3.5-Turbo baseline and CLIP-diversity **0.0900** vs. **0.0879**.
> Overall, LPIPS-based diversity remains essentially unchanged (all methods within about 1-2%), while CLIP-diversity changes mildly: it decreases slightly for SDXL (0.0985$\rightarrow$0.0802). This supports our original claim that the VFM discriminator slightly reshapes the distilled generator toward human-preferred semantic modes; and diversity change is small compared to the gains in human-preference-aligned metrics.
>
> **Qualitative visualization.**
> To further illustrate this effect, in Fig. 10 of the revised paper (please check the revised pdf file),  we visualize 16 samples for the prompt “*a teddy bear on a skateboard in Times Square*” for DMD2-SDXL (left) and Ours-SDXL (right). Both models exhibit clear variation in pose, background, and apparel. DMD2-SDXL shows slightly more variation in viewpoint (e.g., more side-angle shots of the bear), while our model tends to produce more centered, compositionally consistent bears—matching the mild CLIP-diversity difference and the improved human-preference performance reported in Tab. 2.
>
> Overall, the new quantitative and qualitative evidence supports our original interpretation: VFM Discriminator slightly reshapes the distribution toward human-preferred, semantically meaningful modes and causes a mild reduction in certain diversity measures. We have added these results and a short clarification to Appendix~B.6 in the revised paper.
>
>
> We report LPIPS-diversity (average pairwise LPIPS distance) and CLIP-diversity (variance in CLIP image embeddings) on a 23-prompt validation set, with 16 samples per prompt. Higher is better.
>
> | **Method**         | **LPIPS-Diversity** $\uparrow$ | **CLIP-Diversity** $\uparrow$ |
> | ------------------ | ------------------------------ | ----------------------------- |
> | DMD2-SDXL          | 0.5960                         | **0.0985**                    |
> | Ours-SDXL          | 0.6002                         | 0.0802                        |
> | SD 3.5 Large Turbo | 0.5659                         | 0.0879                        |
> | Ours SD~3.5 Large  | 0.5664                         | **0.0900**                    |
>
>
> ###  W2.1&Q2.1: IDA on small-scale model
>
> We agree that it is important to verify that IDA is not only helpful for the largest teachers. In fact, IDA was originally developed and validated on a *2B-parameter SD 3.5-medium* model, which is substantially smaller than the SD 3.5-large and FLUX.1 dev used in our main experiments. In the Fig. 7 and Fig. 8 in our paper, we also provided qualitative comparisons on SD 3.5-Medium, illustrating the visual improvements brought by incorporating IDA. In the revision we now make these results explicit: The table below reports "DMD+VFM" vs. "DMD+VFM+IDA" on SD 3.5-medium. Adding IDA consistently improves the model's performance (HPSv2, AESv2, PickScore, ImageReward, etc), indicating that IDA also helps in the small-model regime rather than hurting it.
>
> | **Method**        | **HPSv2** | **AESv2** | **PickScore** | **ImageReward** |
> | ----------------- | -------------------- | -------------------- | ------------------------ | -------------------------- |
> | DMD w/ VFM        | 0.2843               | 5.362                | 22.59                    | 0.9824                     |
> | DMD w/ VFM w/ IDA | **0.2882**           | **5.547**            | **22.63**                | **1.0100**                 |
>
> We did not enable IDA for SDXL in the current submission because, empirically, DMD training on SDXL was already quite stable and did not exhibit the pronounced inner-loop instabilities we observed for other models. Conceptually, IDA is architecture- and scale-agnostic, due to compute budget constraints we focused our ablations on SD~3.5-medium, and we plan to additionally evaluate IDA on other models in future experiments.

---

> ### Author Response · Authors · 2025-11-21
> **Response to Reviewer xdRf (Part II)**
>
> ### W2.2&Q2.2: Choice of anchor timesteps
>
> Thank you for asking about the impact of the timestep/anchor settings in Appendix~B.1. Our design choice is to *reuse the native inference schedule of each teacher* rather than introduce an additional layer of per-model tuning.
>
> Concretely, SDXL uses the 1000-step discrete DDPM schedule from DMD2; we simply select four anchors along this trajectory at indices $\{249, 499, 749, 999\}$. For SD~3.5 (both medium and large), the base model adopts the official timestep-shift parameterization with a fixed shift value $s=3.0$ for $1024^2$ images. We follow exactly this schedule and place our four anchors on the shifted trajectory, so that the student sees the same coarse time locations as the teacher. For FLUX, the teacher uses a *resolution-dependent* time-shift: the shift parameter is obtained from a simple linear function of the image resolution (e.g. between $256^2$ and $4096^2$), and then applied via the standard time\_shift formula. All our FLUX experiments are conducted at $1024^2$ resolution, and we use the corresponding teacher shift value at this resolution when defining the coarse anchors.
>
> Intuitively, these anchors only define a coarse partition of the trajectory; the proposed ISG then adaptively samples intermediate points within each segment. As long as anchors follow the teacher’s own schedule and cover the denoising interval reasonably evenly, we do not expect small changes in the shift or indices to qualitatively change the distilled model’s behavior—such changes would mostly rescale segment lengths rather than introduce new failure modes. A more systematic sweep over different shift values and anchor placements could reveal a performance curve as a function of the schedule, which is an interesting direction but orthogonal to our main contributions; in this work we intentionally keep the schedule simple and aligned with the base models to ensure stable training and avoid additional hyper-parameter search.

---

> ### Author Response · Authors · 2025-11-21
> **Response to Reviewer xdRf (Part III)**
>
> ### W3&Q3: 1-2 step(s) generation
>
> We appreciate the reviewer’s suggestion to evaluate SenseFlow beyond the 4-step regime. Motivated by this comment, we have conducted evaluations for 2-step and 1-step generation, summarized in the table below.
>
> We find that SenseFlow naturally extends to 2-step sampling. Using the same setup and evaluation protocol, our 2-step sampling results consistently outperform strong baselines. On SDXL, our method surpasses DMD2-SDXL on all human-preference metrics (e.g., HPSv2, AESv2, PickScore, ImageReward). On SD 3.5 Large, the improvement is particularly striking: at 2 steps, our method achieves an ImageReward of **1.2022** and a PickScore of **22.88**, clearly above SD 3.5-Large-Turbo. For FLUX, our 2-step result also improves over Hyper-FLUX on all metrics.
>
> For 1-step generation, we start from the pretrained 4-step SenseFlow models and apply a short fine-tuning (6000 iterations) for the single-step setting. This yields competitive performance with only a modest drop from the 2-step case; for example, the 1-step SD 3.5 Large model reaches an ImageReward of **1.0651** and a CLIP Score of **0.3332**. We also include qualitative comparisons for 2-step and 1-step sampling in the revised Appendix (Fig. 11 and Fig. 16). Especially on SD 3.5 Large, the visual degradation from 4 steps to 2 steps is very small, illustrating that SenseFlow remains robust even in aggressive low-NFE regimes.
>
> Overall, these results demonstrate that SenseFlow provides strong generation quality at 2 steps without architectural changes and exhibits promising single-step performance after light tuning, highlighting its potential to further push the frontier of 1-2 step distillation for large (8B+) text-to-image models, where existing open-source solutions are still scarce.
>
>
> We compare our distilled students with corresponding strong baselines for SDXL, SD~3.5, and FLUX.
>
> | **Method**        | **NFE** | **CLIP** $\uparrow$ | **HPSv2** $\uparrow$ | **AESv2** $\uparrow$ | **PickScore** $\uparrow$ | **ImageReward** $\uparrow$ |
> | ----------------- | ------- | ------------------- | -------------------- | -------------------- | ------------------------ | -------------------------- |
> | DMD2-SDXL         | 2       | 0.3295              | 0.2813               | 5.432                | 22.57                    | 0.8666                     |
> | Ours-SDXL         | 2       | 0.3263              | **0.2827**           | **5.710**            | **22.89**                | **0.9192**                 |
> | Ours-SDXL         | 1       | **0.3298**          | 0.2818               | 5.584                | 22.24                    | 0.8570                     |
> | SD3.5-Large-Turbo | 2       | 0.3248              | 0.2745               | 5.394                | 22.16                    | 0.7188                     |
> | Ours-SD3.5 Large  | 2       | 0.3326              | **0.2889**           | **5.537**            | **22.88**                | **1.2022**                 |
> | Ours-SD3.5 Large  | 1       | **0.3332**          | 0.2803               | 5.421                | 22.32                    | 1.0651                     |
> | Hyper-FLUX        | 2       | 0.3176              | 0.2682               | 5.541                | 21.63                    | 0.3636                     |
> | Ours-FLUX         | 2       | **0.3207**          | **0.2866**           | **5.926**            | **22.38**                | **0.9296**                 |

---

> > ### Author Response · Authors · 2025-11-27
> > **Comment by Authors**
> >
> > Thank you for handling our manuscript and providing valuable feedback. We hope that our responses have sufficiently addressed the concerns you raised. We welcome more discussion if you have more questions and suggestions. As the discussion deadline is approaching, we would be very grateful if you could take a moment to review our reply.

---

### Official Review · Reviewer_qo4N · 2025-10-31

**Soundness:** 3
**Presentation:** 3
**Contribution:** 3
**Rating:** 6
**Confidence:** 4

**Summary:**

This paper proposes an improved version of DMD (VSD). The idea is to set the parameters of the fake score model as a linear combination of the fake model and the student model. The paper also proposes to reallocate timestep importance to help convergence.

**Strengths:**

1. The paper studies the problems of DMD and proposes a fix. The proposed method achieves good results for finetuning Flux and SD3.5.

2. The paper proposes good ablation experiments to help readers understand the contribution of each component.

3. The proposed method is faster compared with DMD.

**Weaknesses:**

1. Although I recognize IDA as a meaningful contribution, the proposed method IDA only works when the student model and fake model have the same network structure. For example, VSD (DMD) is originally designed for text-to-3D. Or more broadly speaking, the distribution matching does not require the student model to be same as the teacher model (real model here). In these general settings, the IDA does not work.

2.  The images in Fig5, Ours-SD3.5 seem too bright. Same thing for Fig 10. What could be the cause?

**Questions:**

Please refer to the weaknesses.

---

> ### Author Response · Authors · 2025-11-21
> **Response to Reviewer qo4N**
>
> ### W1: On the architectural assumption of IDA and possible extensions.
>
> We appreciate the reviewer’s comment about the scope of IDA. Our current instantiation indeed assumes that the generator (student) and the fake model share the same backbone, so that the IDA step can be implemented as a simple interpolation in parameter space. This matches the regime we focus on in this paper (distilling large image/video generative models), where in practice the student is almost always initialized from the teacher weights and the fake model is chosen to share this architecture as well.
>
> More conceptually, IDA is designed to address a specific issue that arises in the early phase of DMD training: when the generator has not converged yet, $p_g$ is very poor and moves quickly. In this regime, the samples produced by the generator provide an unreliable target for the fake model, and updating the fake model solely based on these samples can easily push it in the wrong direction. On the other hand, at initialization we have $p_f \approx p_g$ (in particular, at the first iteration $p_f = p_g$, and the generator receives a DMD gradient that is explicitly guided by the teacher, i.e., a relatively stable direction that points from $p_g$ towards the real distribution $p_{\text{real}}$. From this more global viewpoint, one can reinterpret the idea behind IDA as follows: instead of only updating the fake model using gradients induced by the current generator samples, we can also use the same "$p_g \rightarrow p_{\text{real}}$" gradient signal that drives the generator to additionally update the fake model. Intuitively, IDA also acts as an implicit mechanism for passing teacher/real-distribution supervision to the fake model, rather than letting it rely purely on noisy generator-induced gradients. In other words, the fake model update becomes a combination of an unreliable signal coming from the drifting generator samples and a more reliable signal aligned with the teacher-guided direction.
>
> Our parameter-space interpolation is one concrete realization of this intuition when the architectures match, but the above gradient-based view is not tied to any specific network structure and could, in principle, be applied even when the generator and fake model have different architectures (e.g., in text-to-3D setups). We have not explored such architecture-agnostic variants in this work, and we view this as an interesting direction for future research.
>
> ### W2: On the brightness of Ours-SD3.5 samples in Figures 5 and 10.
>
> We appreciate the reviewer’s careful observation. When training the DMD student on SD 3.5 Large, we indeed noticed that, in the early stage, the generator tends to produce over-bright / bluish images (this is also visible in the "Without ISG" side of Fig. 6). Our current opinion is that this comes from the way the DMD gradient is computed: the real model uses CFG sampling, which makes teacher samples contain more semantically relevant information, but also slightly biased towards areas with higher contrast. In the first few thousand iterations, the student can emphasize this guided direction before fully matching the overall distribution, leading to a mild brightness shift. We experimented with reducing the CFG scale, but the qualitative change was limited, and we did not observe the same effect when distilling other backbones (e.g., FLUX, SDXL), suggesting that this behaviour is somewhat specific to the SD 3.5 Large + DMD setting.
>
> In the revised paper, we have added more qualitative comparisons with diverse prompts (see Fig. 16 in the revised paper). These additional visualizations indicate that, in terms of overall brightness and style, Ours-SD3.5 is generally closer to the SD 3.5 Large teacher than to SD 3.5 Turbo, even though a few individual samples can appear slightly brighter than the teacher outputs.

---

> > ### Comment · Reviewer_qo4N · 2025-11-21
> > **Thanks for the rebuttal**
> >
> > After reading the reviews from other reviewers and the author rebuttal, my concerns are addressed.
> > Considering the contributions in the paper, I would like to maintain my rating of 6.

---

> > > ### Author Response · Authors · 2025-11-22
> > > **Thanks for your response**
> > >
> > > Thank you very much for taking the time to read our rebuttal and for your constructive feedback throughout the review process. We really appreciate your thoughtful comments and are glad to hear that your concerns have been addressed.

---

### Author Response · Authors · 2025-11-22
**General Response**

We thank all reviewers for their time and insightful comments, and are encouraged that they acknowledge the effectiveness of SenseFlow in addressing the challenges of scaling DMD to large flow-based models and its strong experimental results [**qo4N, xdRf, j3tb, iTJr**], the novelty and stabilizing effect of IDA [**xdRf, iTJr**], the rich ablations and training efficiency [**qo4N, xdRf**], and the clear writing and overall structure of the paper [**j3tb**].

- **Method.** SenseFlow scales distribution matching to strong flow-based teachers by combining implicit distribution alignment (IDA), intra-segment guidance (ISG), and a VFM-based discriminator. The method is training-friendly for SDXL / SD 3.5 / FLUX and supports very low NFE (2–4 steps, even 1 step) sampling.
- **Experiments.** The method consistently improves standard Benchmarks (GenEval, T2I-Compbench), human-aligned metrics (HPSv2, AESv2, PickScore, ImageReward) and quality metrics across three model families and multiple NFEs.
- **Theory.** We provide a formal perspective on IDA, connecting local tracking of the fake model to a KL gap bound between the generator and teacher.

Meanwhile, we thank all reviewers again for their constructive suggestions that helped us further improve the paper. We summarize the revisions and added analyses as follows:

- [**iTJr**] We clarified the FM-OT example, fixed the typo in the forward process, and revised the assumptions, lemmas, and propositions around IDA to make the guarantees and required conditions explicit.
- [**iTJr**] We clarified the conceptual relation to CTM and other flow-map / MeanFlow based methods. We will explicitly cite and discuss CTM and related work in the revised version.
- [**xdRf, j3tb**] We expanded ablations on SD 3.5 (Medium and Large) and SDXL to disentangle the effects of ISG, IDA, and the VFM discriminator (and the reference image feature in Discriminator), and to show that IDA also brings consistent gains for smaller models.
- [**xdRf**] We added diversity analyses (LPIPS-diversity and CLIP-diversity) and discussion.
- [**xdRf**] We reported additional low-NFE results (1-step and 2-step) on SDXL, SD 3.5, and FLUX.
- [**xdRf, j3tb**] We added more implementation details, including the size of the filtered LAION subset, training iterations and batch statistics, teacher timestep schedules, and time-shift choices.
- [**j3tb, iTJr**] We added richer qualitative side-by-side comparisons for SDXL, SD 3.5 Large, and FLUX (Figures 15–17), where we compare SenseFlow with strong baselines under the same prompts and observe sharper details, cleaner structure, and more faithful overall quality.
- [**j3tb**] We analyzed the behavior of DMD2-style baselines at scale, including the collapse issues we observed on training DMD2 for SD 3.5 Large.
- [**iTJr**] We incorporated a human preference study with ~20 participants over diverse prompts and three model families, together with richer qualitative comparisons.
- [**qo4N**] We clarified the architectural assumption behind IDA and discussed how the underlying gradient-based idea can, in principle, be applied even when the generator and fake model have different architectures. We also clarified the “brightness” observation on SD 3.5 and showed that SenseFlow remains stylistically close to the teacher.

All changes in the new PDF are highlighted for easy reference.

We hope to have addressed all raised concerns and would be happy to respond to further questions and suggestions. Thanks for your time and feedback.

---

### Author Response · Authors · 2025-12-03
**Summary of rebuttal for Area Chair**

We sincerely thank the Area Chair for taking on the extra responsibility and workload resulting from the recent review reset. To facilitate their work under these circumstances, we have prepared this brief overview of the final reviewer positions and the resolutions reached during the discussion period.

The discussion confirmed that three of the four reviewers [**qo4N, xdRf, j3tb**] explicitly stated positive rate. Before the discussion was interrupted, Reviewer **iTJr** stated that most of the concerns had been addressed and expressed a clear intention to increase the score; however, the interruption occurred before iTJr was able to provide a final rating. We have also thoroughly addressed all the concerns. A brief overview is provided in the table below:

| Reviewer | Main Concern                                                | Rebuttal Action / Fix                                                                                   | Last Response                            |
|----------|-------------------------------------------------------------|---------------------------------------------------------------------------------------------------------|------------------------------------------|
| **qo4N** | Architectural assumption of IDA | Clarified the architectural assumption behind IDA and discussed how the gradient-based idea can be applied when generator and fake model are different. | Explicitly stated positive final rating of 6. |
| - | Brightness of the SenseFlow-SD 3.5 Large | Provided additional visualization results to show that SenseFlow remains stylistically close to the teacher.	| - |
| **xdRf** | Diversity study of SenseFlow | Added diversity analyses (LPIPS-diversity and CLIP-diversity) and discussion. | Explicitly positive original rating of 6. |
| - | IDA on small-scale model      | Expanded ablations to show that IDA also brings consistent gains on small-scale models.	| - |
| - | Choice of anchor timesteps | Clarified the motivation of anchor timesteps' choices. | - |
| - | 1-2 step(s) generation | Reported additional low-NFE results (1-step and 2-step) on SDXL, SD 3.5, and FLUX. | - |
| **j3tb** | ISG’s impact on training stability and convergence speed. | Provided the "FID - Training Iteration" curves to show the improvement of ISG on training stability and convergence speed. | Explicitly stated positive final rating of 6. |
| - | Design and motivation of the VFM discriminator. | Clarified the motivation of adding the reference image feature in discriminator and provided ablation results to show its improvement. | - |
| - | Number of training images. | Reported the size of the filterd training LAION dataset. | - |
| - | Compare with DMD2 | Analyzed the behavior of DMD2-style baselines at scale, including the collapse issues we observed on training DMD2 for SD 3.5 Large. | - |
| - | More ablation variants of Table 2. | Expanded ablations on SD 3.5 Large and SDXL to disentangle the effects of ISG, IDA, and the VFM discriminator. | - |
| **iTJr** | Clarification on Eq. (4) and the roles of $X_0$/$X_1$. | Clarified the FM-OT example, fixed the typo in the forward process. | Expressed a clear intention to increase the score. |
| - | Definition of $v_\theta(X_t,t)$ | Clarified the formal definition of $v_\theta(X_t,t)$ and evised Appendix A to remove the ambiguity of the previous “induces a velocity field” phrasing.	| - |
| - | Role of IDA in Proposition A.7. | Clarified the role of IDA and how it interacts with Proposition A.7, revised the assumptions and propositions around IDA. | - |
| - | Relation to CTM and other flow–map based methods. | Clarified the conceptual relation and difference with CTM and other flow-map / MeanFlow based methods. | - |
| - | The proof of Lemma A.3 | Clarified the field Lipschitz in Assumption A.1 to support Lemma A.3 | - |
| - | Human Evaluation. | Incorporated a human preference study, together with richer qualitative comparisons. | - |
| - | TTUR(10) appears more unstable in Fig. 3 | Clarified the non-monotonic behavior of TTUR and explained why TTUR(10) becomes unstable. | - |

We hope this summary aids in the final decision-making process.

---

### Meta-Review · Area_Chair_yjdZ · 2026-01-04

**Summary:**

This paper focuses on step distillation for text-to-image diffusion models. Specifically, SenseFlow is introduced in this paper, which scales distribution matching to strong flow-based teachers by combining implicit distribution alignment (IDA), intra-segment guidance (ISG), and a VFM-based discriminator. The proposed method theoretically and empirically addresses the covergence difficulties of DMD and achieves good performance on several benchmarks. In the initial review, the paper is rated as 6,6,6, and 4. The main concerns lie in algorithm application and explanation, comparison with previous works, more ablation studies and writing. During rebuttal, authors provides detailed explanation and more experiments to demonstrate the effectiveness of the paper. Three reviewers explicitly keep the positive rating and the remaining reviewer's concern are mostly addressed. Therefore, AC recommends this paper as **Accept (Poster)**.

**Reviewer Concerns:**

The initial main concerns lie in algorithm application and explanation, comparison with previous works, more ablation studies and writing.

Algorithm application and explanation: authors clarified the design and assumption of the key components, and provides more implementation details. This concern is mostly addressed.

Comparison with previous works: authors provides the comparison with DMD2, SD3.5-Large-Turbo and Hyper-FLUX in different NFE settings. This concern is mostly addressed.

More ablation studies: results on small-scale model, as well as the ablation studies on effects of ISG, IDA, and the VFM discriminator are presented. This concern is mostly addressed.

Writing: authors revise the paper, fixing the typos and clarify some paragraphs.

During discussion, Reviewer iTJr raised a question about the definition of velocity field. AC confirms that this is a basic definition in flow matching models and the authors have also explained this to Reviewer iTJr in their further response.

**Reviewer Scores:**

**Reviewer qo4N** keeps 6. Reviewer qo4N replied to the rebuttal and stated to keep the rating.

**Reviewer xdRf** keeps 6. Reviewer xdRf did not have the chance to reply to the rebuttal, but AC confirms that the concerns are mostly addressed.

**Reviewer j3tb** keeps 6. Reviewer j3tb replied to the rebuttal and stated to keep the rating.

**Reviewer iTJr** increases to 6. Reviewer iTJr replied to the rebuttal and mentioned that most of the concerns are addressed except for the theoretical components. Reviewer iTJr expressed an intention to increase the rating if the authors address the concerns regarding theoretical components. AC confirms that the further response from authors addressed this concern.

---

### Decision · Program_Chairs · 2026-01-26

Accept (Poster)